# ProtoLLM: Training and Example-free LLMs for Few-shot Tabular Learning

## Abstract

Recent breakthroughs in large language models (LLMs) have opened the door to in-depth investigation of their potential in tabular data modeling. However, the paradigm for effectively utilizing advanced LLMs in few-shot and even unseen scenarios remains to be explored. We observed an unusual phenomenon: directly using LLMs for data augmentation or rule generation by feeding a few examples significantly degrades the reasoning ability in tabular data understanding. We identified two main obstacles behind this issue: overfitting to the examples and knowledge disruption. Specifically, the provided examples may introduce noisy patterns that interfere with the model's prior knowledge, leading to unexpected and less reliable results. To this end, we propose an example-free framework to leverage the inherent knowledge of LLMs. Our key idea is to prompt the LLM for feature value generation based solely on task and feature description. Without such example pollution, each output feature value is treated as a standard guideline, and they together act as a prototype for each class. To transfer the LLM's knowledge to a given task, we further design an efficient fusion strategy to integrate the prototype with examples, showing impressive generalizability in the few-shot setting. Importantly, our pipeline requires no learnable variables, resulting in a desired training-free property. Extensive comparisons and ablations on multiple tabular datasets demonstrate the improvements of our simple framework.

## 1 Introduction

Large Language Models (LLMs) have shown impressive understanding abilities for solving unseen tasks. Functional as both knowledge repositories and reasoning engines, they are often viewed as the holy grail in recent machine learning fields (Achiam et al., 2023; Badaro et al., 2023; Dubey et al., 2024). This sparks a research trend that focuses on applying LLMs to tabular data analysis. Tabular data, which consists of structured rows and columns (or samples and features), is a critical format in industries such as finance (Arun et al., 2016; Clements et al., 2020), healthcare (Ulmer et al., 2020; Zhou et al., 2020), and more (Buczak & Guven, 2015; Guo et al., 2017). Unlike textual sequences, the features in tabular data are often heterogeneous, and their interrelationships are not inherently sequential. This complexity poses significant challenges for tabular data learning, particularly in real-world few-shot constraints, where algorithms are expected to exploit prior knowledge to learn effectively from limited labeled samples. In contrast, traditional algorithms often struggle to optimize in such constrained settings (Ucar et al., 2021; Han et al., 2024).

To tackle such knowledge deficit issues, recent research has focused on integrating tabular learning into language generation pipelines, leveraging LLMs to enhance few-shot performance. Pre-trained in extensive datasets with vast parameters, LLMs embody a rich repository of prior knowledge and demonstrate a near-human level of comprehension (Brown et al., 2020; Bommasani et al., 2021; Chen et al., 2023; Touvron et al., 2023; Kong et al., 2024). Notably, they can quickly adapt to new tasks through task-oriented prompts. This prompt-based approach is not only user-friendly but also significantly enhances the model's ability to transfer pre-trained knowledge to novel and unseen scenarios efficiently. In alignment with these principles, recent attempts have designed prompt templates such as "*<Meta-Info>-<Example>-<Query>*", where *<Meta-Info>*, *<Example>* and *<Query>* denote the task and feature descriptions, few-shot examples, and user queries, respectively, outperforming conventional tabular learning baselines in few-shot regimes. For instance, TabLLM (Hegselmann et al., 2023) evaluates several ways of serializing tabular examples into nat-

Figure 1: Example-based and example-free outputs of LLMs for feature inference. Example-based LLMs exhibit overfitting tendencies, while example-free LLMs provide more generalized outputs.

ural language strings, facilitating the efficient interpretation of structured inputs by LLMs. Besides, FeatLLM (Han et al., 2024) demonstrates that LLMs can be viewed as rule generators and prompts LLMs to directly output decision rules for each class by feeding the few-shot examples, showing a more efficient strategy to utilize prior knowledge.

While these example-based LLMs have achieved impressive results, we observe an unusual phenomenon: poor-quality examples could degrade the reasoning ability of LLMs. In few-shot learning, the provided examples typically serve as an empirical approximation of the true data distribution; unfortunately, they often fail to capture the full complexity and variability of underlying patterns, particularly in the context of heterogeneous tabular features (Harari & Katz, 2022; Jin et al., 2023; Li et al., 2024b). Moreover, due to the limited number of examples, these samples may contain spurious correlations or causality, introducing irrelevant features or misleading guidance. As shown in Figure. 1, example-based LLMs attempt to learn from the given examples, they may overfit to biased patterns, which could conflict with their pretrained knowledge, leading to inaccurate predictions and undermined the ability to generalize effectively to new, unseen data. Similar challenges have also been observed in recent studies in the fields of computer vision and natural language processing (Li et al., 2024a; Liu et al., 2024).

To address above overfitting and knowledge pollution issues, in this paper, we propose ProtoLLM, which is an example-free framework to employ LLMs as prototype builders for zero and few-shot tabular data classification. Intuitively, as collecting high-quality examples is often nontrivial in practice, we ask whether one can draw out the task-specific knowledge of LLMs with example-free inputs. We are inspired by recent success in zero-shot learning literature, which queries LLMs only with task descriptions and demonstrates that LLMs are decent zero-shot reasoners (Kojima et al., 2022; Wang et al., 2023). Specifically, we remove the <Example> term from the prompt template and encourage LLMs to generate feature values according to their inherent knowledge about the given tasks. This simple strategy avoids the introduction of poor-quality examples, allowing LLMs as human experts to think using task-specific common sense, and thus predict the reasonable value "*high*" in the case of Figure 1. We highlight the following advantages of the proposed ProtoLLM:

- **Training and Example-Free**. Applying ProtoLLM to tabular data classification requires no learnable variable. More importantly, querying ProtoLLM is zero-shot. LLMs take the prompt "<Meta-Info>-<Query>" as input and output the discriminative feature values for each class, resulting in a novel example-free framework for tabular data learning. We further propose to build the class-level prototype by combining the few-shot examples and generated feature values from LLMs in the observed space. Therefore, we can classify the test sample directly by comparing its Euclidean distance with the prototype of each class, making ours a training-free method.

- **Feature Value Generation by LLM**. Unlike traditional data augmentations that employ LLMs to generate all features of a tabular sample simultaneously, we focus on querying LLMs feature by feature. This feature-level generation relieves LLMs from the complex inter-feature relationships, resulting in a meaningful feature discovery. Furthermore, this trick can also act as a novel data augmentation tool after a simple concatenation along the

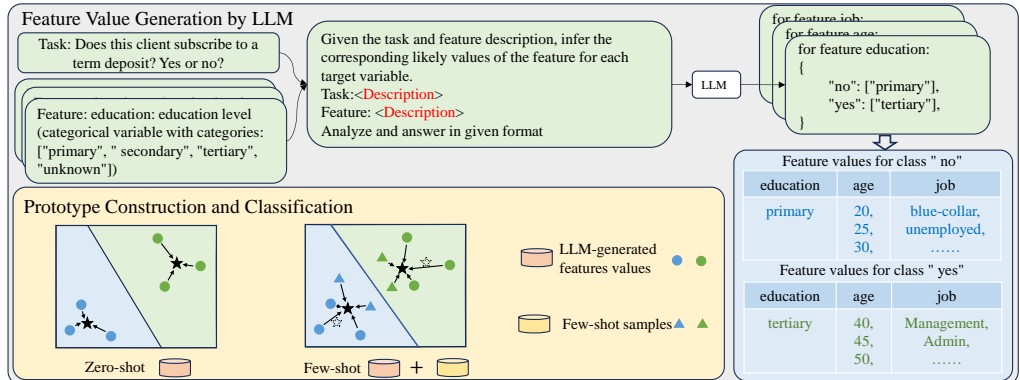

Figure 2: Overview of our proposed ProtoLLM. The upper part shows our example-free feature value generation by LLMs, which predicts the potential values of the feature for each class by feeding our customized prompt into LLMs. Here the prompt depends on the description about task and feature from the dataset. The lower part shows our prototype construction and classification, which builds the class-level prototype with the generated feature value by LLM and introduces a training-free classification framework.

feature dimension. We find the generated features can improve previous baselines significantly (Fig. 7).

- **Prior Knowledge and Target Examples Disentanglement**. Thanks to the above advantages, if we have a few examples, we can shift the prior feature values derived from LLMs to the target domain through a fusing strategy. The resulting feature values are viewed as combined prototypes, and they benefit from both of the LLMs' knowledge and few-shot domain guidance. As the prior feature values are obtained via an example-free manner, such disentanglement can mitigate the misalignment between LLMs and examples. This helps to utilize the examples more efficiently, showing promising results in few-shot settings (Fig. 5).

We compare ProtoLLM with advanced baselines on multiple tabular datasets in zero and few-shot settings and provide detailed ablations from various perspectives, showing its robust and superior performance. By generating discriminative values for each feature, ProtoLLM outperforms FeatLLM by a clear margin, providing a solid option for future studies on tabular data analysis.

## 2 RELATED WORK

**Few-shot Tabular Data Learning**. The development of effective algorithms for few-shot tabular learning has long been a popular research topic, due to their wide variety of applications (Borisov et al., 2022; Kadra et al., 2021; Sattarov et al., 2023). Inspired by the great success across various modalities (Schick & Schütze, 2020; Wang et al., 2024), previous works have proposed a number of few-shot learning frameworks using different techniques, including Bayesian inference, data augmentation, self-supervised learning and others. For example, TabPFN (Hollmann et al., 2023) design a prior to model complex uncertainty of tabular data and show promising performance in small tabular classification tasks. STUNT (Nam et al., 2023) uses unlabeled data to generate diverse few-shot tasks, demonstrating that meta-learning is an effective solution for tabular data, while some works show that contrastive learning is another option to learn general features (Verma et al., 2021; Shenkar & Wolf, 2022). As a follow-up, a series of self-supervised learning methods are proposed for tabular data, they often adopt data augmentation trick and reconstruction loss to extract useful patterns (Majmundar et al., 2022; Ucar et al., 2021; Bahri et al., 2022).

**Understanding Tabular Data with LLMs**. The impressive performance of LLMs has highlighted their broad knowledge and potential in instruction following and low-shot understanding (Ouyang et al., 2022; Brown et al., 2020). This motivated recent attempts to integrate LLMs with tabular data learning (Manikandan et al., 2023; Hollmann et al., 2024; Dinh et al., 2022; Wen et al., 2024). For instance, Curated LLM (Seedat et al., 2023) leverages the in-context capabilities of LLMs for data augmentation and shows that LLMs are high-quality tabular data generators. TabLLM (Hegselmann et al., 2023) takes tabular examples as input and fine-tunes LLMs on few-shot tabular datasets,

showing competitive results with traditional tree-based models. P2T (Nam et al., 2024) propose a transfer learning framework based on LLM that facilitates classification with a limited amount of labeled data. To avoid end-to-end prediction and fine-tuning in utilizing LLMs, FeatLLM (Han et al., 2024) employs LLMs to extract rules for better prediction while achieving relatively low inference time. Our ProtoLLM is most relevant to these LLM-based models but differs from them in terms of feature generation and label prediction.

## 3 METHODS

This work introduces ProtoLLM, a novel training-free and example-free framework for integrating LLMs into tabular few-shot classification. The overview of ProtoLLM is shown in Fig. 2. The core idea behind ProtoLLM is to generate discriminative feature values by querying LLMs with example-free prompts, as described in Sec. 3.1. This avoids knowledge pollution in previous example-based models and ensures the output features come solely from LLMs. Given the generated values of each feature, we design a training-free method to build prototype and classify new samples, as described in Sec. 3.2. Ours provides a promising solution to use LLMs and examples more efficiently.

**Problem Formulation.** We consider a dataset with $N$ samples, denoted as $\mathcal{S} = \{(\boldsymbol{x}_n, y_n)\}_{n=1}^{N}$, where $N$ is usually small in low-shot tasks and $N = 0$ means the zero-shot setting. Each sample $\boldsymbol{x}_n$ consists of $D$ features in total. $x_n^d$ denotes the $d$-th feature of $\boldsymbol{x}_n$, which can be either a numerical feature or a categorical feature. Specifically, if $x_n^d$ is a numerical feature, then $x_n^d \in \mathbb{R}$ and represents a scalar value. If $x_n^d$ is a categorical feature, it is represented as a one-hot encoded vector with 1 denoting the corresponding category. The label $y_n \in \{1, \ldots, C\}$ indicates the class of the sample, with $C$ being the total number of classes. Denote $\mathcal{F} = \{f_{\text{task}}, f_{\text{feat}}^{1:D}\}$ as the set of descriptive information for the dataset, which are usually available in tabular dataset. Specifically, $f_{\text{task}}$ is the information related to the task and $f_{\text{feat}}^d$ is the descriptions corresponding to the $d$-th feature. Taking the Adult dataset as the example, "education" is the $d$-th feature, $f_{\text{feat}}^d$ now includes the explanation about the "education" itself, e.g. education level, and its feature values, e.g. primary, secondary, tertiary, unknown. Learning from $\mathcal{S}$, ProtoLLM aims to predict the label of new data.

### 3.1 FEATURE VALUE GENERATION BY LLM

To draw out the prior knowledge of LLMs and enable them to analyze our problem like a human expert, as illustrated in Fig.3, we carefully design an example-free prompt by proposing a novel way. Specifically, for $d$-th feature, we denote the prompt as $P(f_{\text{task}}, f_{\text{feat}}^d) = [\langle Meta\text{-}Info\rangle\text{-}\langle Query\rangle]$, where we shorten it to $P_d$ for convenience. The prompt is started with "you are an expert in ...", a classic and shared sentence in prompt engineering. Then, we design "$\langle Meta\text{-}Info\rangle$" by introducing the information about task and the $d$-th feature. It is used to provide basic information into LLMs. Besides, "$\langle Query\rangle$" is constructed with a reasoning instruction followed by a requirement of output formation. For the $d$-th feature, we query LLMs with the prompt template $P_d$ and expect that LLMs output correct feature values for each class. We describe this process in more detail below.

**Task and Feature Information in $\langle Meta\text{-}Info\rangle$.** For LLMs, it's crucial to provide a clear task description and detailed feature information. The task description explains the objective, scope, and expected outcomes. As shown in Fig.3, we summarize the $\langle Meta\text{-}Info\rangle$ in red words. For the Adult dataset, the task is "*Does this person earn more than 50000 dollars per year? Yes or no?*". Each feature description outlines the input variables used in prediction, clarifying their roles. For instance, "relationship" refers to the individual's family role, such as spouse or child, which can impact income potential. The description is "*What this individual is relative to others*". Designed in this way, the meta-information (task and feature descriptions) allows LLMs to understand the current task and leverage their prior knowledge for generating representative feature values.

**Reasoning Instruction & Response Format in $\langle Query\rangle$.** The objective of the prompt is to guide the LLM in generating possible values for each target class based on a given feature. Motivated by the recent chain-of-thought (CoT) tricks (Wei et al., 2022b; Lyu et al., 2023), we construct the reasoning instruction with two steps, i.e., green words in Fig.3. First, we ask LLMs to analyze the potential causality of the task and feature based on the provided information in "$\langle Meta\text{-}Info\rangle$". This allows LLMs to mobilize their pre-trained knowledge about the question. Then, given the output analysis, LLMs are encouraged to infer the possible feature values for each class, where we devise the response format with blue words in Fig.3.

You are an expert in analyzing relationships between features and target variables.
Given a feature description and a task, your goal is to analyze how the feature relates to the target and then generate a dictionary with specific details.

Task: Does this person earn more than 50000 dollars per year? Yes or no?
Feature: relationship, what this individual is relative to others(categorical variable with categories: ['Own-child', 'Husband', 'Not-in-family', 'Unmarried', 'Wife', 'Other-relative'])

First, thoroughly analyze the relationship between the feature and task using your prior knowledge.
Then based on this analysis, create a dictionary with the following format:

The output should be a markdown code snippet formatted in the following schema, including the leading and trailing "```json" and "```":
```json
{
    "no": [list of strings]  // possible values of feature relationship for target class no
    "yes": [list of strings]  // possible values of feature relationship for target class yes
}
```

The relationship values should be presented as strings in lists. And make sure the list of each target class is not empty.

Figure 3: An example prompt of our ProtoLLM to generate feature values for the "relationship" attribute in the Adult dataset. Here, black words mean the classic sentence in prompt, red words denote the descriptions about task and feature, i.e., *<Meta-Info>*. Besides, green words denote the reasoning instruction and blue words denote the response format that we expect LLMs to output, which construct the *<Query>*.

Based on prior knowledge, the relationship feature describes what an individual is relative to others. It is a categorical variable with the following categories: ['Own-child', 'Husband', 'Not-in-family', 'Unmarried', 'Wife', 'Other-relative'].

To analyze the relationship between this feature and the task of earning more than $50,000 per year, we can examine the potential relationship categories that are more likely to have a positive or negative impact on the target variable.

According to common knowledge, individuals who are married or have a spouse (categories: 'Husband', 'Wife') are more likely to have higher household incomes. On the other hand, individuals who are children (category: 'Own-child') or have no family attachments (categories: 'Not-in-family', 'Unmarried', 'Other-relative') might have lower household incomes.

Based on this analysis, we can create the following dictionary with the specific details:
```json
{
    "no": ["Own-child", "Not-in-family", "Unmarried", "Other-relative"],
    "yes": ["Husband", "Wife"]
}
```

Please note that this analysis is based on general knowledge and should be further validated with the dataset at hand.

Figure 4: Response for the "relationship" attribute in the Adult dataset, generated by GPT-3.5, where "no" and "yes" in blue color denote the target class in Adult dataset (whether this person earn more than 50000 dollars per year?). Besides, the red words mean the corresponding discriminative feature values generated by LLMs.

## 3.2 BUILD PROTOTYPE AND CLASSIFY

Given the representative values for each feature, how to utilize them for tabular zero-shot and few-shot classification tasks is a key problem. Different from most of existing methods that use the augmented samples by LLM to train a classifier, we introduce a training-free method by building prototype for each class directly. Let $z_{c,d}$ denote the generated value for $d$-th feature in class $c$:

$$
\begin{cases}
z_{c,d} = \text{LLM}(P_d)[c] & \text{if } d\text{-th feature is a numerical feature,} \\
z_{c,d} = \text{One-hot}(\text{LLM}(P_d)[c]) & \text{if } d\text{-th feature is a categorical feature,}
\end{cases}
\tag{1}
$$

where $P_d$ denotes the prompt input for $d$-th feature, and $\text{LLM}(\cdot)[c]$ denotes the output values of LLMs for $c$-th class. We directly use the output values of LLMs for numerical features and post-process the categorical features with the One-shot$(\cdot)$ function to convert the output class index to a one-hot vector[1]. Considering the robustness, we can query LLMs $K$ times for each feature independently, resulting in a set of feature values: $\mathcal{Z}_{c,d} = \{z_{c,d}^k\}_{k=1}^K$. To complete the prototype $\Theta_c$ for class $c$, we adopt an average strategy on $\mathcal{Z}$ as follows:

$$
\Theta_c = [\boldsymbol{\theta}_{c,1}, ..., \boldsymbol{\theta}_{c,d}, ..., \boldsymbol{\theta}_{c,D}], \quad \boldsymbol{\theta}_{c,d} = \frac{1}{K} \sum_{k=1}^K z_{c,d}^k,
\tag{2}
$$

where we concatenate all averaged features to build the final prototype $\Theta_c$. Note that this prototype is inferred solely from LLMs via our example-free prompt, and it thus implicitly encodes common knowledge of LLMs for the target task. From the Bayesian perspective, the prototype acts as a domain expert and provides meaningful priors for prototype learning.

Importantly, the prototype in Eq. 2 is built in the zero-shot setting, and it can be simply shifted to the target domain if the few-shot samples are given (we still use $\boldsymbol{\theta}_{c,d}$ for simply):

$$
\boldsymbol{\theta}_{c,d} = \frac{1}{K + |\mathcal{S}_c|} \left( \sum_{k=1}^K z_{c,d}^k + \sum_{\boldsymbol{x}_n \in \mathcal{S}_c} x_n^d \right),
\tag{3}
$$

where $\mathcal{S}_c$ is the subset of $\mathcal{S}$ containing samples with label $c$. The first prior term focuses on general knowledge from LLMs, which presents the common sense of the given task. The second term can be explained as the data likelihood. It contains the domain information encoded in the input samples. Eq. 3 receives information from two different domains and combines them via an average operation. This simple yet efficient fusing strategy helps the pre-trained knowledge transfer to the target distribution, improving the prototype learning in the few-shot setting.

**Prediction.** Once the prototype is calculated, one can predict the label $y$ for a new sample $\boldsymbol{x}$:

$$
p(y = c|\boldsymbol{x}) = \frac{\exp(-\text{Dist}(\Theta_c, \boldsymbol{x})/\tau)}{\sum_{c'=1}^C \exp(-\text{Dist}(\Theta_{c'}, \boldsymbol{x})/\tau)},
\tag{4}
$$

where $\tau$ is a hyper-parameter and $\text{Dist}(\cdot, \cdot)$ denotes the distance between prototype and sample, which is specified as the Euclidean distance by default. Notably, Eqs. 2- 4 are calculated without any learnable variables, resulting in a training-free framework for tabular data classification. We summarize the workflow of ProtoLLM at Algorithm 1 of Appendix.

### 3.3 FURTHER ANALYSIS

**Feature-level Prior Generation.** Here, we analyze the proposed example-free strategy in more depth. Firstly, ProtoLLM generates features in a zero-shot manner. This not only satisfies the practical need for tabular data but also avoids knowledge pollution, providing an efficient solution to utilize the prior knowledge of LLMs. Secondly, ProtoLLM focuses on a single feature at each query time. Unlike previous works that generate all features (or important features) directly, our feature-level strategy transforms the complex reasoning problem into $D$ tractable sub-problems, allowing LLMs to highlight the correlations between the target label and current feature, showing promising presentation learning compared to sample-level methods (Tab. 1). Lastly, the generated features can also be used as augmented samples. As discussed above, these LLM-generated feature values

---

[1]For the case where LLMs output $m$ feature values, $z$ is obtained as: $z = \frac{1}{m} \sum_m z^m$.

capture common properties for each class, and we empirically find that they together can be used as high-quality samples to improve various baselines (Fig. 7).

**Bayesian-aware Prototype Construction.** Generally, Eq. 3 calculates prototypes by explicitly combining the prior knowledge and domain information, which is quite different from previous example-based LLMs. Specifically, those methods feed examples into prompts and expect LLMs to act as Eq. 3 to directly generate the prototype. This implicit strategy, unfortunately, fails to infer the correct prototype when the given poor-quality examples conflict with LLMs' knowledge. In contrast, our ProtoLLM ensures the prior is reasonable in most cases, which could mitigate the knowledge pollution issue and result in better prototypes.

## 4 EXPERIMENTS

In this section, we first introduce the setups of the low-shot tabular data classification and implementation details. Then we evaluate the proposed model with recent advances, including traditional machine learning methods, few-shot algorithms, and LLM-based frameworks. We also conduct extensive ablations to test the effectiveness of the proposed modules.

### 4.1 EXPERIMENTAL SETUPS

**Datasets.** Following the previous FeatLLM, we here focus on few-shot tabular data classification across 10 datasets, including binary or multi-class classification tasks. Specifically, we use Adult (Kohavi et al., 1996), Bank (Moro et al., 2014), Blood (Yeh et al., 2009), Car (Kadra et al., 2021), Credit-g (Kadra et al., 2021), Diabetes (Smith et al., 1988), Heart (Detrano et al., 1989), Myocardial (Golovenkin et al., 2020), and two other datasets including Cultivars (de Oliveira et al., 2023) and NHANES, which were released recently and not included in the LLMs pre-training stage. These datasets cover fields such as financial, medical, and recommendation, varying in size and complexity. Each dataset contains the corresponding name and description for each attribute, which serves as meta-information in our example-free prompt. We summarize all datasets in Appendix A.2.

**Baselines.** We compare ProtoLLM against three types of baselines: *(1)* Traditional machine learning methods. This category includes Logistic Regression (LogReg), K-Nearest Neighbors (KNN), and XGBoost, representing a range of commonly used algorithms and showing robust results in tabular data analysis. *(2)* Deep learning methods. We here examine methods specifically designed for tabular few-shot learning, including STUNT (Nam et al., 2023), TabPFN (Hollmann et al., 2023), and a 2-layer MLP for comparison. *(3)* LLM-based framework. This includes methods such as In-context Learning (Wei et al., 2022a), TABLET (Slack & Singh, 2023), TabLLM (Hegselmann et al., 2023), and FeatLLM (Han et al., 2024). Specifically, we serialize tabular data in a manner similar to TabLLM, but we query GPT-3.5 to ensure a fair comparison with our method in a zero-shot scenario. All of which utilize LLMs' prior knowledge to enhance performance in few-shot tabular data prediction. These LLM-based models are most closely related to our ProtoLLM. Generally, they focus on leveraging LLMs by feeding examples, we aim to explore an example-free framework to draw out priors, avoiding the mentioned knowledge pollution and overfitting issues.

**Implementation Details.** Following FeatLLM, we select *gpt-3.5-turbo-0613* as our base LLM by default. The number of query times $K$ is set to 10 for each feature across all datasets and the temperature $\tau$ in Eq. 4 is fixed at 1. For the baseline models, the results for STUNT, In-context Learning, TABLET, TabLLM, and FeatLLM are derived from Han et al. (2024). We set the number of samples per class as the same as query number $K$ in k-Nearest Neighbors (KNN), and use a hidden dimension of 1024 for Multi-Layer Perceptron (MLP). Implementation details for other baselines can be found in the Appendix A.4.

### 4.2 RESULTS

#### 4.2.1 ZERO AND FEW-SHOT CLASSIFICATION

To evaluate the capability of our ProtoLLM in processing tabular data, we compare it with baselines on the zero and few-shot classification tasks. Specifically, we run ProtoLLM 15 times, only with different seeds, and report the average area under the receiver operating characteristic curve (AUC)

at Fig. 5. We compare the performance of 11 models under various shot settings (4, 8, 16, 32, and 64), with AUC curves displayed in the subfigures for each dataset. Besides, we also report zero-shot results of our ProtoLLM and TabLLM to test the performance of applying LLMs on unseen tabular data classification without example instructions. Due to limitations in some baseline models, we were unable to evaluate them on certain datasets and shot settings. These limitations are explained in Appendix A.5. The overall average rank across 10 datasets is shown in the top-left subfigure, with all numerical results provided in Table 9 of appendix.

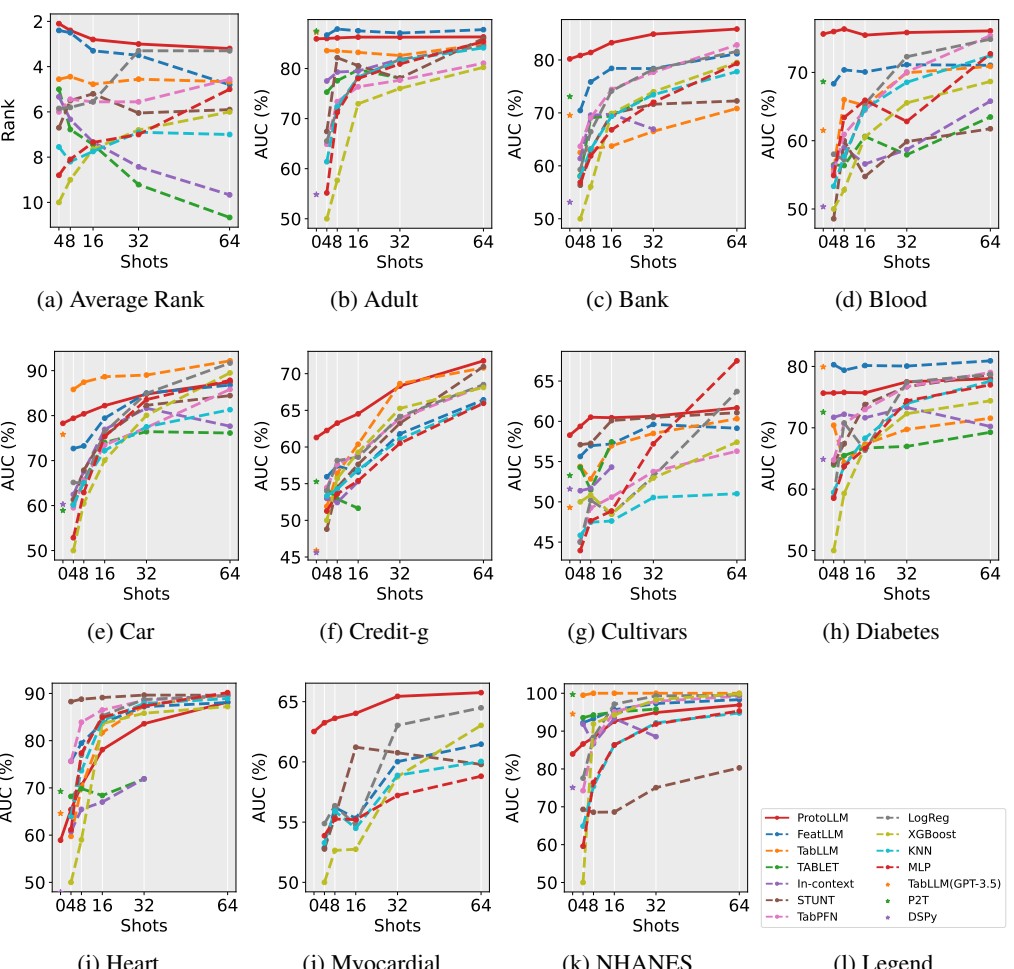

Figure 5: Comparison of AUC scores with different methods across 10 datasets. (a) denotes the average performance on all datasets. (b)-(k) denotes the performance on each dataset, respectively.

From Fig. 5, we have the following remarkable findings. Firstly, our ProtoLLM outperforms existing approaches for most datasets and achieves the No.1 average rank over all zero and few-shot settings. This significantly demonstrates the robustness and adaptability of the proposed ProtoLLM across different settings. Secondly, we find that LLM-based models beat traditional machine learning and deep learning methods with a large gap, especially in the very few-shot settings. There is no doubt that LLMs provide rich domain knowledge and help models recognize the core patterns among heterogeneous features. Furthermore, our ProtoLLM achieves the highest AUC scores on 5/10 datasets compared to the LLM-based models, while in contrast, TabLLM and FeatLLM obtained the highest AUC scores on 2/10 datasets, respectively. We attribute this superiority to the effectiveness of our example-free prototype generation framework. For one thing, the example-free prompt tends to draw out the clean prior knowledge of LLMs, providing more correct features for downstream tasks. For another thing, our simple feature fusing strategy efficiently combines the LLMs prior and data likelihood, resulting in better prototype learning. Lastly, our approach outperforms TabLLM at most zero-shot settings, setting a solid baseline for applying LLM for unseen tabular data classification.

## 4.3 Further Discussion

Given the strong performance on both zero and few-shot settings, we next aim to dig out which components of ProtoLLM play a central role in prompting LLM in tabular data learning. To get the final prototypes, ProtoLLM designs two novel strategies: example-free prompts and feature-level generation. The former ensures a clean output of LLMs solely based on their common sense, and the latter helps LLMs ignore complex correlations between tabular features and focus on the true causality of a given feature and the target label. Consequently, we introduce three variants of ProtoLLM and conduct extensive ablations. Specifically, For the first factor, we modify our example-free prompt by inserting few-shot tabular examples. For the feature-level factor, we ask LLMs to generate all $D$ feature values simultaneously. We report the ablation results in Tab. 1.

We find consistent improvements for ProtoLLM when using feature-level generation without examples in most cases. More interesting, compared to sample-level generation, our approach demonstrates superior performance by fusing the intrinsic characteristics of individual features, which facilitates the construction of more generalized and robust prototypes. Another key observation is that adding examples to both feature-level and sample-level generation led to a noticeable decrease in performance, indicating the presence of example pollution.

Table 1: AUC scores under varying shot settings, compared with different generation methods.

| Data | example | generation | Shot | | | |
|---|---|---|---|---|---|---|
| | | | 0 | 4 | 8 | 16 |
| Adult | ✓ | sample-level | - | 83.99±3.13 | 83.73±2.31 | 85.24±1.97 |
| | | featurel-level | - | 79.44±4.98 | 83.47±1.52 | 84.26±2.95 |
| | | sample-level | 83.59±2.15 | 84.46±1.76 | 84.88±1.54 | 85.19±1.73 |
| | | featurel-level | 85.93±0.64 | 86.01±0.78 | 86.12±0.92 | 86.28±0.77 |
| Bank | ✓ | sample-level | - | 64.53±12.01 | 72.49±7.45 | 74.45±7.90 |
| | | featurel-level | - | 70.28±6.72 | 71.01±5.62 | 76.85±3.82 |
| | | sample-level | 68.80±5.67 | 71.47±5.50 | 72.40±5.19 | 75.75±3.92 |
| | | featurel-level | 80.20±2.22 | 80.85±2.58 | 81.41±2.58 | 83.26±1.40 |
| Blood | ✓ | sample-level | - | 62.84±12.01 | 68.25±9.42 | 66.89±11.10 |
| | | featurel-level | - | 62.93±12.96 | 64.39±10.31 | 68.46±10.02 |
| | | sample-level | 71.60±5.49 | 71.48±5.32 | 71.65±5.02 | 71.33±4.50 |
| | | featurel-level | 75.63±4.15 | 75.98±4.99 | 76.35±4.61 | 75.46±4.12 |
| Cultivars | ✓ | sample-level | - | 46.47±7.92 | 49.29±9.74 | 48.84±8.76 |
| | | featurel-level | - | 47.71±9.54 | 49.62±9.96 | 50.07±9.20 |
| | | sample-level | 52.31±9.47 | 48.94±8.68 | 50.73±8.19 | 50.37±6.31 |
| | | featurel-level | 58.93±8.13 | 59.37±7.98 | 60.51±8.00 | 60.45±7.13 |

**Impacts of Distance Metrics.** In previous experiments, we used the Euclidean distance to calculate the semantic difference between the prototype and the test data. Here, we selected different distance metrics (Euclidean, Manhattan, and Cosine) to evaluate their impact on performance. We report the average AUC score for 10 datasets in Tab.2. From the table, we can observe that the differences in AUC between the various distance methods are relatively small. This insensitivity to the distance function suggests that our generated prototypes capture distinctive features unique to each class and are robust to different semantic metrics. For a more detailed comparison, please refer to Appendix A.9.

Table 2: Mean AUC across all datasets for different distance metrics.

| Shot | Euclidean | Manhattan | Cosine |
|---|---|---|---|
| 0 | 72.07 | 72 | 72.1 |
| 4 | 73.48 | 73.45 | 73.52 |
| 8 | 74.63 | 74.28 | 74.57 |
| 16 | 76 | 75.65 | 76.35 |
| 32 | 78 | 77.06 | 78.36 |
| 64 | 79.8 | 78.46 | 79.9 |

**Impacts of Base LLMs.** Note that our ProtoLLM is LLM-agnostic, which means we can apply various LLMs to improve the tabular data analysis. In this experiment, we compare the performance of the GPT-3.5 and GPT-4o models in zero-shot scenarios to this end. The results are summarized in Tab.3. We find that GPT-4o demonstrates slightly better performance than GPT-3.5 overall, indicating its improved capability in zero-shot understanding tasks. However, in some datasets, GPT-3.5 still maintained competitive performance. We also perform experiments with open-source LLMs to further validate our approach. For details, please refer to Appendix A.10.

**Analysis of Query Times per Feature.** We query LLMs $K$ times for better feature value generation in the previous experiments. We here test the impact of the number of query times and report the average results on 10 datasets in Fig. 6. Generally, $K$ balances the weights of prior knowledge and

data likelihood. As $K$ increases, LLMs become more significant in the final prototype construction. One can obtain higher results after selecting the optimal $K$ on the validation datasets.

Table 3: Comparison of AUC results between GPT-3.5 and GPT-4o in zero-shot scenarios.

| Data | GPT-3.5 | GPT-4o |
|---|---|---|
| Adult | **85.93±0.64** | 85.74±0.51 |
| Bank | 80.20±2.22 | **82.25±0.91** |
| Blood | 75.63±4.15 | **77.83±4.53** |
| Car | **78.29±1.72** | 77.86±1.56 |
| Credit-g | **61.29±3.03** | 58.66±4.62 |
| Cultivars | 58.93±8.13 | **61.16±5.58** |
| Diabetes | **75.65±3.05** | 73.17±3.57 |
| Heart | 58.28±4.30 | **74.37±4.09** |
| NHANES | 83.96±3.21 | **86.38±2.25** |

Table 4: AUC results of ProtoLLM and its feature-weighting variant on four datasets.

| Data | Shot | ProtoLLM | ProtoLLM+weight |
|---|---|---|---|
| Bank | 0 | 80.20 | **81.08** |
| | 4 | 80.85 | **81.06** |
| | 8 | **81.41** | 81.05 |
| Credit-g | 0 | 61.29 | **64.09** |
| | 4 | 62.25 | **65.71** |
| | 8 | 63.26 | **66.63** |
| Diabetes | 0 | 75.65 | **81.55** |
| | 4 | 75.68 | **81.63** |
| | 8 | 75.76 | **81.61** |
| NHANES | 0 | 83.96 | **97.52** |
| | 4 | 86.60 | **98.18** |
| | 8 | 88.30 | **98.44** |

**ProtoLLM as Data Augmentation.** Previous results suggest that our training and example-free framework have great potential for high-quality generation. Let $\hat{x}_c^k = \left[z_{c,1}^k, z_{c,2}^k, \ldots, z_{c,D}^k\right]$ denote $k$-th augmented sample for class $c$. To evaluate such data augmentation abilities of ProtoLLM, we first use these $K \times C$ samples to augment $\mathcal{D}$ and then apply traditional machine learning methods. The results in Fig.7 show that our augmented samples significantly improve the classification accuracy on Logits Regression, K-Nereast Neighbors, and MLP, verifying the superior quality of the data generated by our approach. However, despite these gains, the results still do not surpass the performance of our method, further demonstrating the advantage of the feature-level prototype generation in our approach.

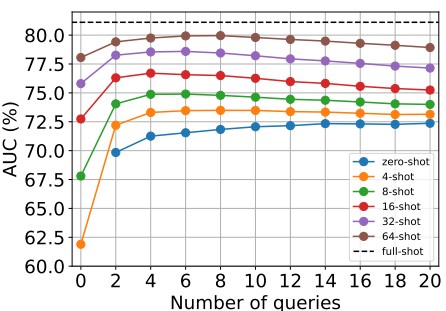

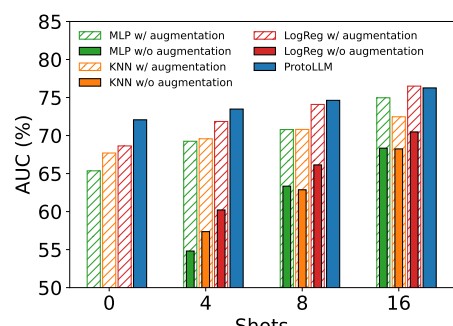

Figure 6: Averaged AUC results of ProtoLLM with various query times.

Figure 7: Average AUC results of baselines with various data augmentations.

**Weighted Feature Generation.** Feature importance is a technique that identifies which features in a tabular dataset significantly influence a model's predictions. Intuitively, considering the feature weights at the prototype construction stage should enhance prediction accuracy. Motivated by this, instead of only generating the feature values, we in this section ask LLMs to simultaneously output feature weights. Tab. 4 illustrates the comparisons of ProtoLLM and its feature-weighting variant. We find ProtoLLM+weight outperforms ProtoLLM significantly in most cases. Detail information is provided in appendix A.12.

## 5 CONCLUSION

We propose ProtoLLM, a training and example-free framework for zero and few-shot tabular data classification. This provides a novel strategy to combine prior knowledge of LLMs and data likelihood for prototype construction. We show that it is possible to efficiently draw out clean common sense of LLMs and generate feature values by designing example-free prompts, and that doing so effectively avoids knowledge pollution and overfitting issues in previous example-based LLMs. Finally, we also showcase our ProtoLLM can be used as data augmentation and boosts traditional algorithms. A number of comparisons and ablations on 10 datasets demonstrate the superior performance of our approach. Our ProtoLLM is LLM-agnostic and can be benefit from stronger LLM. We hope that our approach provide valuable insights into the utility of LLMs for tabular data analysis.

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

# A APPENDIX

## A.1 PROTOLLM WORKFLOW

**Algorithm**

---

**Algorithm 1** Workflow of our proposed ProtoLLM.

---

1: **Require:** Dataset $\mathcal{S} = \{(\boldsymbol{x}_n, y_n)\}_{n=1}^N$ for few-shot and $\mathcal{S} = \emptyset$ for zero-shot, descriptive information $\mathcal{F} = \{f_{\text{task}}, f_{\text{feat}}^{1:D}\}$ about dataset $\mathcal{S}$, test sample $\boldsymbol{x}$, LLM;
2: **Output:** Predicted probability vector $\boldsymbol{p}$ for test sample;
3: **Step 1:** Generate $\mathcal{Z}$ by LLM with our designed example-free prompt and initial $\mathcal{Z} = \emptyset$ as:
4: **for** $d = 1$ to $D$ **do**
5:     Design prompt as $P(f_{\text{task}}, f_{\text{feat}}^d) = [\textit{<Meta-Info>-<Query>}]$, abbreviated as $P_d$;
6:     **for** $k = 1$ to $K$ **do**
7:         Prompt LLM with $P_d$ and output $z_{c,d}^k$ with Eq. 1; set $\mathcal{Z} = [\mathcal{Z}, z_{c,d}^k]$;
8:     **end for**
9: **end for**
10: **Step 2:** Build prototypes $\Theta_{1:C}$ with $\mathcal{Z}$ and $\mathcal{S}$ in Eq. 2 for zero-shot or Eq. 3 for few-shot;
11: **Step 3:** Compute the predicted probability vector $\boldsymbol{p}$ for test sample $\boldsymbol{x}$ with Eq. 4;
12: **return** $\boldsymbol{p}$                                ▷ Return predicted probabilities

---

## A.2 DATASETS AND BASELINES

**Datasets** In this section, we present an overview of the datasets utilized in the experiments. Each dataset is specifically designed for a distinct classification task, with various features employed to predict categorical outcomes. A summary of the key characteristics and classification objectives of each dataset is provided below:

- The **Adult**[2] dataset is used to determine whether an individual earns more than $50,000 annually, based on demographic and employment features.

- The **Bank**[3] dataset predicts whether a customer will subscribe to a term deposit, utilizing personal and socio-economic factors.

- The **Blood**[4] dataset is designed to forecast whether a person will donate blood, given past donation records.

- The **Car**[5] dataset classifies the acceptability of a car based on attributes like buying price, maintenance cost, and safety features.

- The **Credit-g**[6] dataset addresses the classification of individuals as good or bad credit risks, using personal and financial attributes.

- The **Cultivars**[7] dataset assesses the growth and yield of forty soybean cultivars under varying conditions, with features such as plant height, number of stems, and grain yield.

- The **Diabetes**[8] dataset focuses on predicting the likelihood of diabetes in a person, relying on medical metrics such as glucose levels and BMI.

- The **Heart**[9] dataset identifies whether a patient is at risk of heart disease, considering factors like age, cholesterol levels, and blood pressure.

---

[2]archive.ics.uci.edu/dataset/2
[3]archive.ics.uci.edu/dataset/222
[4]archive.ics.uci.edu/dataset/176
[5]archive.ics.uci.edu/dataset/19
[6]archive.ics.uci.edu/dataset/144
[7]archive.ics.uci.edu/dataset/913
[8]kaggle.com/datasets/uciml/pima-indians-diabetes-database
[9]kaggle.com/datasets/fedesoriano/heart-failure-prediction

- The **Myocardial**[10] dataset is used to predict the outcomes of patients following a myocardial infarction, based on clinical data such as heart rate and blood pressure.
- The **NHANES**[11] dataset is derived from the National Health and Nutrition Examination Survey, focusing on predicting respondents' age using features such as physiological measurements, lifestyle factors, and biochemical markers.

Table 5: Properties about all of the datasets.

|  | Adult | Bank | Blood | Car | Credit-g | Cultivars | Diabetes | Heart | Myocardial | NHANES |
|---|---|---|---|---|---|---|---|---|---|---|
| Objects | 48842 | 45211 | 748 | 1728 | 1000 | 320 | 768 | 918 | 1700 | 6287 |
| Numerical | 7 | 8 | 4 | 1 | 8 | 7 | 8 | 7 | 17 | 7 |
| Categorical | 7 | 8 | 0 | 5 | 12 | 3 | 0 | 4 | 94 | 1 |
| Class | 2 | 2 | 2 | 4 | 2 | 2 | 2 | 2 | 2 | 2 |

### A.3 EXAMPLE OF PROMPT AND ANSWER

For numerical features, the prompts differ slightly. We require LLMs to infer their ranges. Additionally, we limit the length of the list due to the variations across different queries. The following example demonstrates a prompt used to generate feature values for the 'age' attribute in the Adult dataset:

You are an expert in analyzing relationships between feature and target variables.
Given a feature description and a task, your goal is to analyze how the feature relates to the target and then generate a dictionary with specific details.

Task: Does this person earn more than 50000 dollars per year? Yes or no?

Feature: age, the age of an individual(age>16)(numeric variable, you should use your prior knowledge to determine the appropriate ranges of values)

First conduct a thorough analysis of the relationship between the feature and task using your prior knowledge.
Then based on this analysis, create a dictionary with the following format:

The output should be a markdown code snippet formatted in the following schema, including the leading and trailing "```json" and "```":

```json
{
"no": [list of integers] // possible values of feature age for target class no
"yes": [list of integers] // possible values of feature age for target class yes
}
```
The age values should be presented as integers in lists. Make sure to include 5 typical age values for each target class in ["no", "yes"].

Figure 8: Prompt for querying values of feature 'age' in Adult dataset.

---

[10]archive.ics.uci.edu/dataset/579
[11]archive.ics.uci.edu/dataset/887

The Answer values for feature 'age' is detailed below, where LLMs analyze the relationship and provide responses for various values of the 'age' feature corresponding to each class in the Adult dataset.

---

Based on prior knowledge, we can analyze the relationship between age and the task of determining if a person earns more than 50000 dollars per year.

Typically, as an individual progresses in their career, their income tends to increase with age until they reach their peak earning potential. Therefore, it is reasonable to expect that people in older age groups are more likely to earn more than 50000 dollars per year compared to younger age groups.

However, it is important to note that this is a general trend and there will always be exceptions. Other factors such as education level, occupation, and work experience can also impact an individual's income.

Based on this analysis, let's create a dictionary with the specified format:

```json
{
"no": [17, 18, 19, 20, 21],
"yes": [35, 40, 45, 50, 55]
}
```

This dictionary includes 5 typical age values for each target class, where "no" represents individuals earning less than or equal to 50000 dollars per year, and "yes" represents individuals earning more than 50000 dollars per year. The age values provided are just examples and can be adjusted based on the specific dataset and its distribution.

---

Figure 9: Answer for querying values of feature 'age' in Adult dataset.

### A.4 BASELINE DETAILS

In this section, we provide a detailed configuration of the experiments conducted.

For each random seed, 20% of the datasets are designated as the test set. We then perform balanced sampling of $K$ instances from the remaining data, following Hegselmann et al. (2023) and Han et al. (2024).

The baseline models of KNN, LogReg, and MLP are implemented using the scikit-learn library, while XGBoost is implemented using its corresponding package. we utilize the number of samples in a class to determine the parameter `n_neighbors` for KNN while utilizing Euclidean distance (with `p=2`) for distance computation. For LogReg, MLP, and XGboost, we employ a four-fold cross-validation approach along with grid search to identify the optimal hyperparameters, if the sample size is sufficient. The hyperparameter space for LogReg is presented in Table 6. For the MLP, we configure `early_stopping` to True, set `n_iter_no_change` to 5, use `'adam'` as the solver, specify `hidden_layer_size` as 1024, and limit `max_iter` to 200. The other hyperparameter space for MLP is detailed in Table 7. For XGBoost, hyperparameter space of XGBoost is shown in Table 8. Considering TabPFN, we use the official GitHub repository with the default parameters.

Table 6: Hyperparameter search space for LogReg.

| Parameter | Search space |
|---|---|
| penalty | {l1, l2} |
| C | 100, {10, 1, 1e-1, 1e-2, 1e-3, 1e-4, 1e-5} |

Table 7: Hyperparameter search space for MLP.

| Parameter | Search space |
|---|---|
| alpha | {1e-3,5e-3,1e-2} |
| learning_rate_init | {1e-4,5e-4,1e-3,5e-3,1e-2} |

Table 8: Hyperparameter search space for XGBoost.

| Parameter | Search space |
|---|---|
| max depth | {2, 4, 6, 8, 10, 12} |
| alpha | {1e-8, 1e-7, 1e-6, 1e-5, 1e-4, 1e-3, 1e-2, 1e-1, 1} |
| lambda | {1e-8, 1e-7, 1e-6, 1e-5, 1e-4, 1e-3, 1e-2, 1e-1, 1} |
| eta | {0.01, 0.03, 0.1, 0.3} |

A.5   MAIN RESULTS

**Detailed main results in the few-shot scenario.** We present the complete results from Figure 5 in Table 9, including the AUC values and their corresponding standard deviations. The "N/A" entries in the table are due to limitations in certain LLM-based frameworks, including In-context, TABLET, and TabLLM, which are restricted by the maximum number of input tokens they can process. This constraint makes it challenging, or even infeasible, to evaluate these baselines in scenarios with a higher number of shots or more complex features. Additionally, TabPFN cannot be applied to datasets with a large number of features, which also contributes to the "N/A" entries. In contrast, our framework effectively handles scenarios with more shots or complex features, as our example-free prompt is designed to efficiently query the LLM for each feature individually

Table 9: AUC across 10 datasets in few-shot scenarios. Bold highlights the highest score, while underline marks the second highest.

| Data | Shot | LogReg | XGBoost | KNN | MLP | TabPFN | STUNT | In-context | TABLET | TabLLM | FeatLLM | ProtoLLM |
|---|---|---|---|---|---|---|---|---|---|---|---|---|
| Adult | 4 | 65.52±12.63 | 50.00±0.00 | 61.39±8.23 | 55.16±13.97 | 64.88±9.82 | 67.43±29.61 | 77.51±5.24 | 75.29±12.24 | 83.57±2.69 | 86.68±0.86 | 86.01±0.78 |
|  | 8 | 71.90±9.16 | 57.68±6.80 | 72.43±4.79 | 71.24±8.13 | 73.39±5.23 | 82.16±6.93 | 79.30±2.89 | 77.56±7.56 | 83.52±4.30 | 87.89±0.06 | 86.12±0.92 |
|  | 16 | 78.27±7.46 | 72.96±4.79 | 78.25±3.01 | 78.01±9.19 | 76.35±4.39 | 80.57±10.93 | 79.50±4.57 | 79.74±5.64 | 83.23±2.45 | 87.54±0.50 | 86.28±0.77 |
|  | 32 | 81.82±5.23 | 76.02±3.39 | 81.59±2.28 | 80.91±6.56 | 77.67±3.00 | 78.08±15.15 | 81.89±4.04 | 78.08±6.70 | 82.60±4.14 | 87.09±0.58 | 86.26±0.71 |
|  | 64 | 84.54±2.97 | 80.24±2.77 | 84.14±1.32 | 85.39±1.90 | 81.07±1.90 | 86.01±0.16 | N/A | N/A | 84.88±0.97 | 87.77±0.31 | 86.32±0.85 |
| Bank | 4 | 59.29±9.86 | 50.00±0.00 | 57.96±4.82 | 56.87±8.76 | 63.68±6.92 | 56.34±12.82 | 61.38±1.30 | 58.11±6.29 | 62.51±8.95 | 70.45±3.69 | 80.85±2.58 |
|  | 8 | 66.46±12.23 | 56.05±9.29 | 63.13±5.90 | 61.86±10.23 | 69.07±7.09 | 63.01±8.78 | 69.57±13.35 | 69.08±6.00 | 63.19±5.79 | 75.85±6.66 | 81.41±2.58 |
|  | 16 | 74.15±6.95 | 69.86±7.85 | 69.38±4.44 | 66.80±10.31 | 74.43±5.15 | 69.85±0.95 | 69.76±8.55 | 69.40±11.28 | 63.73±6.43 | 78.41±1.08 | 83.26±1.40 |
|  | 32 | 78.25±4.29 | 73.99±3.94 | 73.43±4.80 | 71.99±7.52 | 77.68±3.82 | 71.64±1.65 | 66.93±5.67 | 73.61±9.28 | 66.51±3.92 | 78.37±4.50 | 84.88±1.71 |
|  | 64 | 81.61±3.19 | 79.53±3.23 | 77.80±3.34 | 79.35±4.30 | 82.84±2.52 | 72.26±1.62 | N/A | N/A | 70.83±3.43 | 81.18±6.17 | 85.84±1.28 |
| Blood | 4 | 58.02±13.35 | 50.00±0.00 | 53.33±7.87 | 54.90±16.37 | 55.33±15.09 | 48.57±6.04 | 56.30±12.43 | 56.45±15.45 | 55.87±13.49 | 68.34±7.48 | 75.98±4.99 |
|  | 8 | 57.20±11.28 | 52.79±8.66 | 57.73±6.13 | 63.40±9.44 | 60.93±8.75 | 60.00±4.84 | 58.99±10.12 | 56.37±11.56 | 56.01±9.25 | 70.37±3.23 | 76.35±4.61 |
|  | 16 | 65.41±11.70 | 60.55±9.23 | 64.68±10.04 | 65.92±11.80 | 64.49±8.31 | 54.76±4.53 | 56.59±5.21 | 60.62±4.13 | 65.14±7.55 | 70.07±5.19 | 75.46±4.12 |
|  | 32 | 72.30±9.21 | 65.54±8.31 | 68.55±7.22 | 62.82±15.64 | 70.06±8.30 | 59.87±3.72 | 58.69±1.53 | 57.94±4.16 | 69.95±3.39 | 71.13±4.38 | 75.84±4.39 |
|  | 64 | 74.86±6.68 | 68.67±5.44 | 72.46±5.33 | 72.72±7.34 | 75.39±4.40 | 61.75±2.19 | 65.79±2.05 | 63.47±7.36 | 70.88±1.58 | 71.04±4.36 | 76.08±4.51 |
| Car | 4 | 65.14±5.53 | 50.00±0.00 | 60.14±4.74 | 52.85±5.47 | 59.51±6.06 | 61.32±3.83 | 62.47±2.47 | 60.21±4.81 | 85.82±3.65 | 72.69±1.52 | 79.41±1.92 |
|  | 8 | 65.05±7.13 | 60.48±4.56 | 65.02±2.93 | 62.90±8.01 | 65.66±5.29 | 67.86±0.49 | 67.57±3.44 | 65.53±8.00 | 87.43±2.56 | 73.26±1.46 | 80.40±2.04 |
|  | 16 | 76.33±2.45 | 70.07±5.35 | 72.23±2.12 | 75.34±5.22 | 73.47±4.13 | 75.56±2.88 | 76.94±3.04 | 74.02±1.01 | 88.65±2.63 | 79.43±1.24 | 82.22±2.05 |
|  | 32 | 84.95±2.61 | 80.06±3.53 | 77.50±2.94 | 83.56±3.11 | 77.35±3.15 | 82.29±2.34 | 81.64±2.52 | 76.44±4.02 | 89.02±1.50 | 85.01±1.36 | 84.78±1.81 |
|  | 64 | 91.69±2.49 | 89.50±2.82 | 81.32±2.25 | 87.86±3.73 | 85.84±2.83 | 84.45±1.69 | 77.65±3.74 | 76.13±1.17 | 92.18±0.47 | 86.78±0.90 | 87.45±2.01 |
| Credit-g | 4 | 54.01±5.42 | 50.00±0.00 | 53.33±3.87 | 51.27±6.47 | 54.47±4.80 | 48.80±6.76 | 52.99±4.08 | 54.33±6.54 | 51.90±9.40 | 55.94±1.10 | 62.25±2.86 |
|  | 8 | 58.15±7.63 | 55.71±4.87 | 54.26±4.91 | 53.60±7.88 | 57.64±4.79 | 54.50±8.25 | 52.43±4.36 | 52.90±5.79 | 56.42±12.89 | 57.42±3.10 | 63.26±2.87 |
|  | 16 | 58.62±7.92 | 59.28±5.04 | 56.89±5.62 | 55.39±7.83 | 59.02±5.60 | 57.63±7.58 | 55.29±4.80 | 51.65±4.02 | 60.38±14.03 | 56.60±2.22 | 64.52±3.28 |
|  | 32 | 64.16±5.27 | 65.26±4.49 | 61.06±3.53 | 60.50±7.59 | 63.90±4.23 | 63.24±5.47 | N/A | N/A | 68.64±3.86 | 61.79±10.25 | 68.32±3.05 |
|  | 64 | 68.51±5.27 | 68.12±3.51 | 66.03±3.70 | 65.96±8.45 | 68.33±2.90 | 70.97±4.95 | N/A | N/A | 70.80±4.09 | 66.43±2.90 | 71.75±3.31 |
| Cultivars | 4 | 44.98±7.89 | 50.00±0.00 | 45.84±7.29 | 43.95±7.55 | 45.09±6.53 | 57.10±8.66 | 51.38±2.48 | 54.28±3.73 | 54.39±5.61 | 55.63±5.24 | 59.37±7.98 |
|  | 8 | 50.20±8.63 | 50.86±8.09 | 47.47±8.93 | 47.64±10.07 | 49.23±7.58 | 57.26±9.52 | 51.68±4.43 | 51.48±3.85 | 52.86±6.13 | 56.97±5.08 | 60.51±8.00 |
|  | 16 | 48.48±7.76 | 48.48±9.09 | 47.62±9.58 | 48.86±10.01 | 50.58±8.21 | 60.09±7.64 | 54.31±6.12 | 57.44±3.53 | 56.97±2.22 | 57.19±5.30 | 60.45±7.13 |
|  | 32 | 53.15±8.63 | 52.96±7.40 | 50.54±10.05 | 57.20±10.26 | 53.76±9.93 | 60.48±6.51 | N/A | N/A | 58.50±2.65 | 59.62±7.43 | 60.63±6.95 |
|  | 64 | 63.70±9.65 | 57.41±8.20 | 51.01±10.37 | 67.51±6.43 | 56.29±7.26 | 61.07±6.77 | N/A | N/A | 60.32±2.60 | 59.14±4.79 | 61.67±7.62 |
| Diabetes | 4 | 58.74±13.20 | 50.00±0.00 | 59.48±6.89 | 58.53±14.21 | 64.76±10.85 | 64.22±6.78 | 71.71±5.31 | 63.96±3.32 | 70.42±3.69 | 80.28±0.75 | 75.68±3.61 |
|  | 8 | 70.79±5.87 | 59.30±12.01 | 63.70±8.14 | 63.73±8.25 | 70.68±7.87 | 67.39±12.92 | 72.21±2.07 | 65.47±3.95 | 64.30±5.88 | 79.38±1.66 | 75.76±3.78 |
|  | 16 | 66.34±8.61 | 66.88±8.42 | 68.30±5.73 | 66.61±7.82 | 72.98±5.05 | 73.79±6.48 | 71.64±5.05 | 66.71±0.76 | 67.34±2.79 | 80.15±1.35 | 75.70±4.33 |
|  | 32 | 77.44±5.13 | 72.30±4.17 | 73.96±4.70 | 74.39±4.73 | 76.72±4.64 | 76.70±4.55 | 73.32±1.59 | 66.97±1.75 | 69.74±4.41 | 80.06±1.18 | 77.48±3.81 |
|  | 64 | 78.72±3.57 | 74.40±4.10 | 77.59±2.84 | 77.00±6.18 | 79.02±3.49 | 78.64±3.32 | 70.22±4.09 | 69.27±6.15 | 71.56±4.55 | 80.91±1.62 | 78.00±3.86 |
| Heart | 4 | 63.98±19.36 | 50.00±0.00 | 63.98±11.68 | 61.15±18.38 | 75.61±17.34 | 88.27±3.32 | 60.76±4.00 | 68.19±11.17 | 59.74±4.49 | 75.66±4.59 | 65.40±7.18 |
|  | 8 | 76.93±10.21 | 59.00±12.07 | 73.64±11.23 | 77.36±10.24 | 83.91±7.62 | 88.78±2.38 | 65.46±3.77 | 69.85±10.82 | 70.14±7.91 | 79.46±2.16 | 70.55±7.68 |
|  | 16 | 85.27±4.82 | 83.61±5.06 | 84.23±4.08 | 84.91±7.70 | 86.49±4.31 | 89.13±2.10 | 67.00±7.83 | 68.39±11.73 | 81.72±3.92 | 83.71±1.88 | 78.09±5.63 |
|  | 32 | 88.74±2.72 | 85.81±3.83 | 87.88±2.22 | 87.33±5.08 | 88.52±2.75 | 89.65±3.04 | 71.94±3.88 | 71.90±9.07 | 87.43±2.32 | 87.19±3.66 | 83.59±3.88 |
|  | 64 | 89.50±2.19 | 87.21±2.97 | 88.91±1.69 | 90.17±1.85 | 89.63±2.47 | 89.62±3.16 | N/A | N/A | 89.78±2.59 | 88.08±4.11 | 88.17±2.24 |
| Myocardial | 4 | 54.88±8.06 | 50.00±0.00 | 53.28±5.65 | 53.87±9.22 | N/A | 52.77±2.01 | N/A | N/A | N/A | 52.87±3.44 | 63.25±4.16 |
|  | 8 | 56.36±6.18 | 52.65±5.67 | 55.92±6.44 | 55.25±7.53 | N/A | 55.40±4.41 | N/A | N/A | N/A | 56.22±1.64 | 63.62±4.12 |
|  | 16 | 54.77±5.87 | 52.75±7.37 | 54.49±6.75 | 55.21±6.93 | N/A | 61.22±3.45 | N/A | N/A | N/A | 55.32±9.15 | 64.03±4.04 |
|  | 32 | 63.03±6.53 | 58.78±8.04 | 58.88±5.03 | 57.21±7.91 | N/A | 60.76±1.58 | N/A | N/A | N/A | 60.02±4.02 | 65.44±4.38 |
|  | 64 | 64.49±5.90 | 63.03±7.84 | 60.04±3.67 | 58.81±8.67 | N/A | 59.79±0.56 | N/A | N/A | N/A | 61.47±3.91 | 65.75±4.34 |
| NHANES | 4 | 77.56±15.27 | 50.00±0.00 | 64.90±8.82 | 59.58±14.23 | 74.27±11.17 | 69.32±19.59 | 91.84±3.79 | 93.54±4.20 | 99.49±0.23 | 92.20±1.71 | 86.60±3.05 |
|  | 8 | 88.34±12.64 | 91.93±6.99 | 75.38±5.89 | 76.43±6.86 | 87.23±5.82 | 68.56±18.35 | 86.67±5.49 | 94.25±3.35 | 100.00±0.00 | 93.29±7.01 | 88.30±4.34 |
|  | 16 | 97.12±4.35 | 94.25±6.09 | 86.34±4.71 | 86.34±6.05 | 95.12±3.46 | 68.62±19.81 | 93.33±4.47 | 95.02±1.57 | 100.00±0.00 | 95.64±4.67 | 92.61±3.27 |
|  | 32 | 99.30±0.56 | 98.31±2.88 | 94.79±2.16 | 91.88±4.09 | 97.88±1.50 | 75.06±3.56 | 88.54±5.40 | 95.82±3.71 | 100.00±0.00 | 97.29±1.28 | 94.91±2.34 |
|  | 64 | 99.49±0.89 | 99.87±0.47 | 94.79±2.16 | 95.32±2.86 | 99.21±0.66 | 80.29±4.56 | N/A | N/A | 100.00±0.00 | 98.32±0.65 | 96.91±1.40 |
| Average AUC | 4 | 60.21 | 50.00 | 57.36 | 54.81 | N/A | 61.41 | N/A | N/A | N/A | 71.07 | 73.48 |
|  | 8 | 66.14 | 59.65 | 62.87 | 63.34 | N/A | 66.49 | N/A | N/A | N/A | 73.01 | 74.63 |
|  | 16 | 70.48 | 67.87 | 68.24 | 68.34 | N/A | 69.12 | N/A | N/A | N/A | 74.41 | 76.26 |
|  | 32 | 76.31 | 72.90 | 72.56 | 72.78 | N/A | 71.78 | N/A | N/A | N/A | 76.76 | 78.21 |
|  | 64 | 79.71 | 76.80 | 75.41 | 78.01 | N/A | 74.48 | N/A | N/A | N/A | 78.11 | 79.79 |
| Average Rank | 4 | 5.85 | 10.00 | 7.55 | 8.80 | 6.00 | 6.70 | 5.33 | 5.00 | 4.56 | 2.40 | 2.10 |
|  | 8 | 5.80 | 9.00 | 8.20 | 8.10 | 5.44 | 5.50 | 6.33 | 6.78 | 4.44 | 2.50 | 2.40 |
|  | 16 | 5.55 | 7.65 | 7.75 | 7.35 | 5.56 | 5.20 | 7.33 | 7.44 | 4.78 | 3.30 | 2.80 |
|  | 32 | 3.30 | 6.80 | 6.90 | 7.00 | 5.56 | 6.05 | 8.43 | 9.21 | 4.56 | 3.50 | 3.00 |
|  | 64 | 3.30 | 6.00 | 7.00 | 5.00 | 4.56 | 5.90 | 9.67 | 10.67 | 4.67 | 4.80 | 3.20 |

**Detailed main results in the zero-shot scenario.** We compare ProtoLLM against P2T, DSPy, and TabLLM in zero-shot scenarios. To ensure a fair comparison, all baselines leverage GPT-3.5 as the underlying LLM. P2T utilizes additional unlabeled data to facilitate knowledge transfer, while DSPy directly queries the LLM for probability estimates. For TabLLM, we adopt the Text Template method, identified in its original paper as the most effective serialization approach.

Table 10: AUC across 10 datasets in zero-shot scenarios.

|  | DSPy | P2T | TabLLM | ProtoLLM |
|---|---|---|---|---|
| Adult | 54.84±4.24 | **87.49±1.62** | 87.24±0.66 | 85.93±0.64 |
| Bank | 53.14±5.12 | 73.08±9.20 | 69.54±1.79 | **80.20±2.22** |
| Blood | 50.33±3.45 | 68.64±5.62 | 61.51±4.74 | **75.63±4.15** |
| Car | 60.29±2.25 | 58.90±2.17 | 75.80±1.45 | **78.29±1.72** |
| Credit-g | 45.60±2.16 | 55.28±4.01 | 45.88±2.50 | **61.29±3.03** |
| Heart | 47.90±2.80 | **69.27±3.65** | 64.62±3.46 | 58.93±8.13 |
| Diabetes | 64.87±3.10 | 72.55±3.31 | **79.92±2.35** | 75.65±3.05 |
| Cultivars | 51.60±3.60 | 53.27±3.90 | 49.30±1.22 | **58.28±4.30** |
| Myocardial | N/A | N/A | N/A | **62.52±4.48** |
| NHANES | 75.07±1.77 | **99.69±0.24** | 94.53±0.70 | 83.96±3.21 |

## A.6 TOKEN CONSUMPTION

In Table 11, we compare the token consumption per query of our method with that of FeatLLM, which utilizes large language models (LLMs) to extract rules (FeatLLM-rule) and functions (FeatLLM-function). Thanks to its example-free prompt design, ProtoLLM achieves significantly lower token usage compared to both FeatLLM variants, underscoring its superior efficiency.

Table 11: Average number of tokens per query.

|  | Adult | Bank | Blood | Car | Credit-g | Heart | Diabetes | Myocardial | Cultivars | NHANES |
|---|---|---|---|---|---|---|---|---|---|---|
| ProtoLLM | 534.7 | 454.4 | 482.6 | 572.2 | 445.9 | 465.2 | 457.0 | 453.4 | 662.5 | 500.9 |
| FeatLLM-rule | 1254.7 | 903.1 | 765.9 | 610.7 | 869.4 | 888.6 | 784.0 | 1072.6 | 733.143 | 675.571 |
| FeatLLM-function | 3018.7 | 2637.6 | 1320.6 | 1476.6 | 2189.7 | 2347.7 | 2003.7 | 2605.7 | 2270.86 | 2282.86 |

## A.7 FULL-SHOT RESULTS

We present the full-shot results in Table 12. As shown, ProtoLLM under the few-shot setting achieves performance comparable to the full-shot case, with the average performance for the 64-shot setting being marginally different from that of the full-shot case. Moreover, when using LLM-generated feature values on certain datasets (Blood, Cultivars, Myocardial) in the few-shot scenario, the performance even exceeds that of the full-shot case. This highlights the effectiveness of LLM-generated feature values in boosting performance in few-shot scenarios.

Table 12: Performance comparison between full-shot and few-shot settings.

| Dataset | 0-shot | 4-shot | 8-shot | 16-shot | 32-shot | 64-shot | Full-Shot |
|---|---|---|---|---|---|---|---|
| Adult | 85.93±0.64 | 86.01±0.78 | 86.12±0.92 | 86.28±0.77 | 86.26±0.71 | 86.32±0.85 | **87.68±0.32** |
| Bank | 80.20±2.22 | 80.85±2.58 | 81.41±2.58 | 83.26±1.40 | 84.88±1.71 | 85.84±1.28 | **86.86±0.33** |
| Blood | 75.63±4.15 | 75.98±4.99 | **76.35±4.61** | 75.46±4.12 | 75.84±4.39 | 76.08±4.51 | 75.26±4.68 |
| Car | 78.29±1.72 | 79.41±1.92 | 80.40±2.04 | 82.22±2.05 | 84.78±1.81 | 87.45±2.01 | **92.44±1.06** |
| Credit-g | 61.29±3.03 | 62.25±2.86 | 63.26±2.87 | 64.52±3.28 | 68.32±3.05 | 71.75±3.31 | **73.44±3.44** |
| Cultivars | 58.93±8.13 | 59.37±7.98 | 60.51±8.00 | 60.45±7.13 | 60.63±6.95 | **61.67±7.62** | 58.83±7.73 |
| Diabetes | 75.65±3.05 | 75.68±3.61 | 75.76±3.78 | 75.70±4.33 | 77.48±3.81 | 78.00±3.86 | **82.18±2.25** |
| Heart | 58.28±4.30 | 65.40±7.18 | 70.55±7.68 | 78.09±5.63 | 83.59±3.88 | 88.17±2.24 | **90.65±1.84** |
| Myocardial | 62.52±4.48 | 63.25±4.16 | 63.62±4.12 | 64.03±4.04 | 65.44±4.38 | **65.75±4.34** | 65.50±5.17 |
| NHANES | 83.96±3.21 | 86.60±3.05 | 88.30±4.34 | 92.61±3.27 | 94.91±2.34 | 96.91±1.40 | **98.27±0.46** |
| Average | 72.07 | 73.48 | 74.63 | 76.26 | 78.21 | 79.79 | 81.11 |

## A.8 DETAILED RESULTS OF DIFFERENT GENERATION TYPES

Table 13: AUC across 9 datasets with different generation types.

| Data | w/ example | generation-level | Shots 0 | 4 | 8 | 16 |
|---|---|---|---|---|---|---|
| Adult | ✓ | sample-level | - | 83.99±3.13 | 83.73±2.31 | 85.24±1.97 |
| | | featurel-level | - | 79.44±4.98 | 83.47±1.52 | 84.26±2.95 |
| | | sample-level | 83.59±2.15 | 84.46±1.76 | 84.88±1.54 | 85.19±1.73 |
| | | featurel-level | **85.93±0.64** | **86.01±0.78** | **86.12±0.92** | **86.28±0.77** |
| Bank | ✓ | sample-level | - | 64.53±12.01 | 72.49±7.45 | 74.45±7.90 |
| | | featurel-level | - | 70.28±6.72 | 71.01±5.62 | 76.85±3.82 |
| | | sample-level | 68.80±5.67 | 71.47±5.50 | 72.40±5.19 | 75.75±3.92 |
| | | featurel-level | **80.20±2.22** | **80.85±2.58** | **81.41±2.58** | **83.26±1.40** |
| Blood | ✓ | sample-level | - | 62.84±12.01 | 68.25±9.42 | 66.89±11.10 |
| | | featurel-level | - | 62.93±12.96 | 64.39±10.31 | 68.46±10.02 |
| | | sample-level | 71.60±5.49 | 71.48±5.32 | 71.65±5.02 | 71.33±4.50 |
| | | featurel-level | **75.63±4.15** | **75.98±4.99** | **76.35±4.61** | **75.46±4.12** |
| Car | ✓ | sample-level | - | 72.71±4.12 | 74.74±2.91 | 77.51±2.76 |
| | | featurel-level | - | 68.84±5.22 | 72.69±2.96 | 79.37±3.02 |
| | | sample-level | 67.52±4.11 | 69.39±4.34 | 70.81±4.28 | 74.45±4.72 |
| | | featurel-level | **78.29±1.72** | **79.41±1.92** | **80.40±2.04** | **82.22±2.05** |
| Credit-g | ✓ | sample-level | - | 53.10±6.82 | 52.87±8.69 | 59.61±7.08 |
| | | featurel-level | - | 58.11±4.92 | 60.71±3.42 | 60.57±6.10 |
| | | sample-level | 52.80±5.04 | 54.96±6.34 | 55.72±5.11 | 58.53±6.19 |
| | | featurel-level | **61.29±3.03** | **62.25±2.86** | **63.26±2.87** | **64.52±3.28** |
| Cultivars | ✓ | sample-level | - | 46.47±7.92 | 49.29±9.74 | 48.84±8.76 |
| | | featurel-level | - | 47.71±9.54 | 49.62±9.96 | 50.07±9.20 |
| | | sample-level | 52.31±9.47 | 48.94±8.68 | 50.73±8.19 | 50.37±6.31 |
| | | featurel-level | **58.93±8.13** | **59.37±7.98** | **60.51±8.00** | **60.45±7.13** |
| Diabetes | ✓ | sample-level | - | 78.30±3.86 | 77.88±4.33 | 77.71±4.22 |
| | | featurel-level | - | 71.57±6.28 | 71.55±8.03 | 74.48±5.05 |
| | | sample-level | **80.48±2.64** | **80.45±2.84** | **80.41±3.01** | **80.05±3.00** |
| | | featurel-level | 75.65±3.05 | 75.68±3.61 | 75.76±3.78 | 75.70±4.33 |
| Heart | ✓ | sample-level | - | **76.30±10.52** | **84.05±5.94** | **87.90±2.66** |
| | | featurel-level | - | 70.53±18.59 | 80.79±9.61 | 86.47±3.72 |
| | | sample-level | **66.32±8.67** | 75.68±8.74 | 79.70±7.15 | 85.72±3.54 |
| | | featurel-level | 58.28±4.30 | 65.40±7.18 | 70.55±7.68 | 78.09±5.63 |
| NHANES | ✓ | sample-level | - | 85.86±6.82 | **90.06±4.83** | **94.18±2.10** |
| | | featurel-level | - | 76.04±12.39 | 84.66±6.90 | 90.32±2.84 |
| | | sample-level | **85.91±4.01** | **88.08±5.39** | 89.95±4.79 | 93.88±2.07 |
| | | featurel-level | 83.96±3.21 | 86.60±3.05 | 88.30±4.34 | 92.61±3.27 |

## A.9 COMPARISON OF DIFFERENT DISTANCE METRICS

Table 14: AUC across 10 datasets with different distance metrics.

| Data | Shot | Shot | | | | | |
|---|---|---|---|---|---|---|---|
| | | 0 | 4 | 8 | 16 | 32 | 64 |
| Adult | Euclidean | 85.93 | 86.01 | 86.12 | 86.28 | 86.26 | 86.32 |
| | Manhattan | 88.11 | 87.93 | 87.94 | 87.83 | 87.58 | 87.33 |
| | Cosine | 85.85 | 85.88 | 85.91 | 85.97 | 85.83 | 85.88 |
| Bank | Euclidean | 80.2 | 80.85 | 81.41 | 83.26 | 84.88 | 85.84 |
| | Manhattan | 74.03 | 76 | 77.07 | 78.97 | 81.07 | 82.11 |
| | Cosine | 80.18 | 80.84 | 81.39 | 83.32 | 84.95 | 85.91 |
| Blood | Euclidean | 75.63 | 75.98 | 76.35 | 75.46 | 75.84 | 76.08 |
| | Manhattan | 75.73 | 75.24 | 75.02 | 74.61 | 73.51 | 74.01 |
| | Cosine | 75.25 | 76.17 | 75.92 | 76.48 | 76.69 | 76.47 |
| Car | Euclidean | 78.29 | 79.41 | 80.4 | 82.22 | 84.78 | 87.45 |
| | Manhattan | 75.66 | 76.46 | 77.31 | 78.97 | 81.43 | 83.65 |
| | Cosine | 79.29 | 80.27 | 81.13 | 82.7 | 85 | 87.59 |
| Credit-g | Euclidean | 61.29 | 62.25 | 63.26 | 64.52 | 68.32 | 71.75 |
| | Manhattan | 60.98 | 61.8 | 62.83 | 63.99 | 67.45 | 70.37 |
| | Cosine | 60.99 | 61.92 | 62.99 | 64.21 | 68.16 | 71.5 |
| Cultivars | Euclidean | 58.93 | 59.37 | 60.51 | 60.45 | 60.63 | 61.67 |
| | Manhattan | 56.1 | 55.87 | 56.16 | 56.22 | 55.03 | 57.41 |
| | Cosine | 58.92 | 59.22 | 60.67 | 60.26 | 61.17 | 62.12 |
| Diabetes | Euclidean | 75.65 | 75.68 | 75.76 | 75.7 | 77.48 | 78 |
| | Manhattan | 76.3 | 76.71 | 77.08 | 76.65 | 78.01 | 78 |
| | Cosine | 75.74 | 75.97 | 76.21 | 76.27 | 78.17 | 78.4 |
| Heart | Euclidean | 58.28 | 65.4 | 70.55 | 78.09 | 83.59 | 88.17 |
| | Manhattan | 66.74 | 71.98 | 76.02 | 81.83 | 86.15 | 89.37 |
| | Cosine | 58.42 | 65.28 | 70.14 | 77.44 | 83.3 | 88.24 |
| Myocardial | Euclidean | 62.52 | 63.25 | 63.62 | 64.03 | 65.44 | 65.75 |
| | Manhattan | 63.52 | 64.09 | 64.07 | 64.56 | 65.66 | 65.91 |
| | Cosine | 62.29 | 63.08 | 63.16 | 63.77 | 65.03 | 65.5 |
| NHANES | Euclidean | 83.96 | 86.6 | 88.3 | 92.61 | 94.91 | 96.91 |
| | Manhattan | 86.62 | 88.39 | 89.35 | 92.91 | 94.76 | 96.48 |
| | Cosine | 84.04 | 86.54 | 88.2 | 93.04 | 95.31 | 97.42 |

## A.10 COMPARISON OF AUC SCORES WITH DIFFERENT LLMS

We extend our experiments to include open-source LLMs, specifically Llama-3B and Llama-8B, to further validate our approach. Our results show that ProtoLLM continues to achieve comparable performance with these open-source models, demonstrating the robustness of our method across different base LLMs.

Table 15: Comparison of AUC scores with different LLMs.

| dataset | shot | Llama3B | Llama8B | GPT-3.5 | GPT-4o |
|---|---|---|---|---|---|
| Adult | 0 | 80.85±0.35 | 84.78±0.34 | 85.93±0.64 | **85.74±0.51** |
| | 4 | 81.87±1.47 | 84.87±0.48 | **86.01±0.78** | 85.82±0.59 |
| | 8 | 82.76±1.39 | 85.05±0.60 | **86.12±0.92** | 85.91±0.64 |
| | 16 | 83.59±1.18 | 85.10±0.79 | **86.28±0.77** | 85.94±0.60 |
| Blood | 0 | 73.34±3.22 | **77.91±4.42** | 75.63±4.15 | 77.83±4.53 |
| | 4 | 75.16±4.96 | 77.15±4.30 | 75.98±4.99 | **77.17±4.67** |
| | 8 | 76.03±4.89 | **76.99±4.58** | 76.35±4.61 | **76.99±4.72** |
| | 16 | **76.69±4.16** | 76.01±4.25 | 75.46±4.12 | 76.21±4.39 |
| Diabetes | 0 | **79.84±1.77** | 72.67±2.87 | 75.65±3.05 | 73.17±3.57 |
| | 4 | **80.02±1.81** | 73.06±3.34 | 75.68±3.61 | 73.85±3.76 |
| | 8 | **79.95±2.02** | 73.67±3.46 | 75.76±3.78 | 74.16±3.82 |
| | 16 | **79.91±2.26** | 73.60±3.98 | 75.70±4.33 | 74.08±4.48 |
| Heart | 0 | 63.30±4.40 | 64.33±4.22 | 58.28±4.30 | **74.37±4.09** |
| | 4 | 70.79±7.48 | 73.03±8.20 | 65.40±7.18 | **79.30±5.99** |
| | 8 | 75.51±7.63 | 77.95±7.94 | 70.55±7.68 | **82.09±5.68** |
| | 16 | 82.04±5.04 | 84.13±4.75 | 78.09±5.63 | **85.36±3.96** |

## A.11 APPLYING DATA AUGMENTATION TO OTHER MODELS

Table 16: Applying Data Augmentation to LogReg,KNN, and MLP.

| Data | Model | Shot | | | | | |
|------|-------|------|------|------|------|------|------|
| | | 0 | 4 | 8 | 16 | 32 | 64 |
| Adult | LogReg | 76.52±2.57 | 76.83±5.44 | 80.98±5.12 | 82.60±3.39 | 83.73±2.99 | 85.21±1.66 |
| | KNN | 82.79±1.65 | 83.41±1.13 | 83.84±1.07 | 84.14±0.84 | 84.52±0.69 | 85.09±0.59 |
| | MLP | 71.63±9.76 | 80.20±7.03 | 81.45±7.39 | 85.40±3.23 | 85.95±2.51 | 85.43±3.88 |
| Bank | LogReg | 67.83±9.80 | 76.07±5.90 | 78.20±4.46 | 81.51±2.87 | 81.01±4.21 | 84.32±3.52 |
| | KNN | 75.81±2.18 | 76.87±2.40 | 77.17±2.41 | 78.34±1.91 | 80.35±2.16 | 82.44±1.81 |
| | MLP | 65.60±10.36 | 72.18±12.70 | 73.63±8.60 | 78.22±2.51 | 80.92±2.12 | 82.67±2.59 |
| Blood | LogReg | 74.02±4.61 | 73.53±5.50 | 75.40±5.69 | 76.76±4.14 | 76.12±4.89 | 76.76±4.15 |
| | KNN | 73.18±4.83 | 74.41±4.20 | 74.70±3.39 | 74.79±3.56 | 74.75±4.61 | 75.04±4.18 |
| | MLP | 69.32±7.67 | 68.51±9.09 | 67.49±14.09 | 73.94±4.55 | 73.22±7.31 | 73.19±8.12 |
| Car | LogReg | 68.78±4.53 | 72.84±4.69 | 76.79±4.18 | 79.70±3.42 | 82.07±3.04 | 87.67±2.07 |
| | KNN | 72.87±1.68 | 73.93±1.81 | 74.37±1.97 | 75.59±2.07 | 78.88±1.69 | 83.18±1.64 |
| | MLP | 76.17±2.45 | 76.25±2.64 | 77.84±3.30 | 80.69±2.85 | 83.65±2.62 | 87.40±2.54 |
| Credit-g | LogReg | 53.91±4.38 | 57.92±6.76 | 62.64±4.82 | 64.48±5.36 | 68.82±3.98 | 71.27±4.06 |
| | KNN | 59.23±5.60 | 58.48±5.61 | 58.92±4.41 | 61.87±4.59 | 65.96±5.28 | 68.51±4.35 |
| | MLP | 56.17±8.63 | 60.21±6.95 | 63.10±5.82 | 62.51±5.51 | 66.96±4.99 | 69.34±4.70 |
| Cultivars | LogReg | 60.34±8.64 | 51.98±8.57 | 55.99±11.06 | 59.30±9.85 | 62.56±7.83 | 69.55±9.65 |
| | KNN | 53.69±5.90 | 58.65±5.50 | 57.55±5.29 | 54.81±5.00 | 55.19±7.49 | 54.00±8.46 |
| | MLP | 60.81±7.74 | 56.69±8.48 | 58.61±9.41 | 62.70±9.73 | 66.37±8.53 | 73.79±5.30 |
| Diabetes | LogReg | 75.53±5.73 | 76.69±5.34 | 75.14±5.56 | 76.01±4.23 | 77.39±4.96 | 79.78±3.67 |
| | KNN | 71.74±3.53 | 71.70±3.34 | 72.23±3.62 | 73.13±3.31 | 75.50±3.36 | 76.22±3.37 |
| | MLP | 69.17±5.89 | 72.71±6.56 | 66.80±8.89 | 73.76±3.70 | 76.11±3.38 | 75.43±5.96 |
| Heart | LogReg | 50.48±6.67 | 71.32±13.83 | 76.27±11.97 | 83.93±6.55 | 85.90±4.17 | 88.37±2.13 |
| | KNN | 58.55±5.61 | 63.21±9.22 | 68.66±10.98 | 75.88±8.01 | 82.80±5.62 | 88.21±2.91 |
| | MLP | 56.24±11.22 | 63.64±10.55 | 71.38±11.92 | 76.26±10.07 | 85.53±4.76 | 89.42±1.87 |
| Myocardial | LogReg | 60.15±5.29 | 62.57±5.12 | 62.38±5.23 | 63.50±4.97 | 65.34±5.08 | 68.76±4.31 |
| | KNN | 50.00±0.00 | 54.44±5.30 | 57.61±6.37 | 58.36±6.58 | 59.13±5.86 | 60.46±5.10 |
| | MLP | 54.25±8.44 | 61.99±5.16 | 62.67±4.97 | 62.98±3.73 | 65.17±3.94 | 65.00±4.49 |
| NHANES | LogReg | 98.80±1.39 | 98.78±1.48 | 97.17±3.17 | 98.90±0.89 | 99.58±0.47 | 99.86±0.14 |
| | KNN | 79.17±3.45 | 80.62±2.51 | 83.01±3.24 | 87.77±4.07 | 92.34±3.58 | 95.98±2.04 |
| | MLP | 74.25±9.67 | 80.09±5.96 | 84.91±5.55 | 93.21±4.01 | 95.01±3.44 | 98.84±1.06 |

## A.12 DETAIL OF WEIGHTED FEATURE GENERATION

In this section, we outline the prompt utilized to query feature weights for a dataset. For this querying process, both the task and all relevant feature descriptions are provided to the LLMs to ensure a comprehensive understanding of the context and specific characteristics of the data. Additionally, detailed instructions and the required response format are included. Below is an example using the Car dataset.

---

You are an expert in analyzing relationships between features and target variables.
I will provide you with the task description and feature descriptions of a dataset. Your goal is to analyze the importance of each feature in predicting the target variable based on the relationship between features and the target.

Task:How would you rate the decision to buy this car? Unacceptable, acceptable, good or very good?

Feature:
buying: buying price
maint: price of the maintenance
doors: number of doors
persons: capacity in terms of persons to carry
lug_boot: the size of luggage boot
safety: estimated safety of the car

The output should be a markdown code snippet formatted in the following schema, including the leading and trailing "```json" and "```":

```json
{
"buying": float
"maint": float
"doors": float
"persons": float
"lug_boot": float
"safety": float
}
```

Please provide the importance of each feature in predicting the target variable. The importance of each feature should be a float value between 0 and 1.

---

Figure 10: Prompt for querying feature weights in the Car Dataset.

The output is as follows: LLMs first analyze the relationships between features and the target class, then provide a dictionary that delineates the associated feature weights. The generated feature weights could enhance ProtoLLM greatly.

```json
{
"buying": 0.4,
"maint": 0.3,
"doors": 0.1,
"persons": 0.15,
"lug_boot": 0.05,
"safety": 0.6
}
```

Based on the given feature descriptions, the importance of each feature in predicting the target variable can be estimated as follows:

- "buying" (buying price): 0.4 - The buying price of a car can significantly influence the decision to buy. Higher buying prices may indicate better quality or features, leading to a higher rating.

- "maint" (maintenance price): 0.3 - The maintenance price of a car can also play a role in the decision. Lower maintenance costs may indicate affordable upkeep, which can contribute to a positive rating.

- "doors" (number of doors): 0.1 - The number of doors may be a minor factor in the decision to buy. More doors can provide convenience and flexibility, but it may not have a significant impact on the overall rating.

- "persons" (capacity in terms of persons to carry): 0.15 - The capacity to carry more individuals can be an important consideration for families or individuals who frequently travel with a larger group. It contributes to the overall rating but may not be as crucial as other factors.

- "lug_boot"" (size of luggage boot): 0.05 - The size of the luggage boot may have a minimal impact on the decision. While it can be a convenience factor, other features such as safety or maintenance may hold more weight in determining the rating.

- "safety" (estimated safety of the car): 0.6 - The estimated safety of the car is likely to be a significant factor in the decision to buy. Higher safety ratings can greatly influence the overall rating, indicating a higher likelihood of a positive rating.

Figure 11: Generated Feature weights of Car dataset.

Here we present a summary of AUC scores for various datasets using the ProtoLLM model with and without feature weights across different shot configurations (0, 4, and 8). While some datasets show a slight decline in performance with feature weights, many exhibit significant improvements, particularly the NHANES dataset, highlighting the effectiveness of weighted features in enhancing model performance.

Table 17: AUC scores for various datasets using the ProtoLLM model are presented, comparing performance with and without feature weights across different shot configurations.

| Data | Shot | ProtoLLM | ProtoLLM+weight |
|---|---|---|---|
| Adult | 0 | **85.93** | 84.37 |
| | 4 | **86.01** | 84.42 |
| | 8 | **86.12** | 84.56 |
| Bank | 0 | 80.20 | **81.08** |
| | 4 | 80.85 | **81.06** |
| | 8 | **81.41** | 81.05 |
| Blood | 0 | **75.63** | 74.66 |
| | 4 | 75.98 | **76.26** |
| | 8 | 76.35 | 76.35 |
| Car | 0 | **78.29** | 77.73 |
| | 4 | **79.41** | 79.21 |
| | 8 | **80.40** | 80.30 |
| Credit-g | 0 | 61.29 | **64.09** |
| | 4 | 62.25 | **65.71** |
| | 8 | 63.26 | **66.63** |
| Cultivars | 0 | **58.93** | 58.84 |
| | 4 | **59.37** | 59.14 |
| | 8 | 60.51 | **61.91** |
| Diabetes | 0 | 75.65 | **81.55** |
| | 4 | 75.68 | **81.63** |
| | 8 | 75.76 | **81.61** |
| Heart | 0 | **58.28** | 56.61 |
| | 4 | **65.40** | 62.26 |
| | 8 | **70.55** | 66.41 |
| NHANES | 0 | 83.96 | **97.52** |
| | 4 | 86.60 | **98.18** |
| | 8 | 88.30 | **98.44** |
| Average AUC | 0 | 73.13 | **75.16** |
| | 4 | 74.62 | **76.43** |
| | 8 | 75.85 | **77.47** |

## A.13 LIMITATIONS

**Limitations.** This work introduces an example-free and training-free ProtoLLM for zero and few-shot tabular classification tasks. It prompts the LLM for feature value generation based solely on task and feature description, making it difficult to apply to datasets without such describing information. Besides, our ProtoLLM prompts the LLM without the training examples and designs the training-free prototype for classification instead of training a classifier. Therefore, it is more suitable for the zero and extreme few-shot regimes. With the increasing number of training samples, our approach is gradually closer to or inferior to other methods that depends on training samples and a learnable classifier, which is an interesting direction and can be considered in future work.

