# OpenReview forum: "ProtoLLM: Training and Example-free LLMs for Few-shot Tabular Learning"
_ICLR.cc/2025/Conference — Submitted to ICLR 2025_

### Official Review · Reviewer_j7jo · 2024-10-18

**Soundness:** 3
**Presentation:** 2
**Contribution:** 2
**Rating:** 6
**Confidence:** 2

**Summary:**

This paper presents ProtoLLM a novel approach to leverage LLM for supervised classification on tabular data. More precisely, the presented method focuses on few-shot and zero-shot learning for tabular data.

The authors observe that that few-shot learning of LLMs for supervised tasks on tabular data can perform poorly if the few examples fed to the model are not properly selected. The authors thus propose to generate a prototype solely based on the existing knowledge of the LLM through careful prompting, and proceed to use this prototype in a training free maner to compute a prediction for a sample of interest. Their approach can also be extended to few-shot learning by aggregating existing examples with the generated prototype (e.g. averaging the feature values).

The authors test their approach on a tabular data benchmark composed of 10 datasets following previous work, and compare to existing methods. They show that their approach performs strongly based on the AUROC metric and also perform several ablations studies to further demonstrate the relevance of their proposed method.

**Strengths:**

- The proposed method does not require any training and inference is perfomed only based on a distance measure between the sample of interest and the generated prototypes.
- The proposed approach is relevant for both zero-shot and few-shot learning set-ups.
- Proposed approach has (to our knowledge) significant novelty.
- The experiments are extensive and fair, and show that ProtoLLM offers strong performance on the tested benchmark.
- Ablation study is rather exhaustive and investigates most of the relevant factors.
- Work is reproductible.

**Weaknesses:**

- The paper is sometimes hard to follow and wordy. The overall style could be improved, and part of the paper contains either typos or odd turns of phrase (e.g. line 356:"*To validate the seasoning ability of our ProtoLLM on tabular data*", line 300: "*where $\tau$ is a temperature hype-parameter*", line 381: "*We attribute this superior to our example-free prototype generation*".).
- While the proposed approach is novel and interesting, the **contribution is relatively thin** for a conference like ICLR.
- The proposed approach requires prompting an LLM $d \times K \times C$ times, where $d$ is the number of features, $K$ the number of iteration considered to construct the prototype, and $C$ the number of classes. This might restrains this type of approach to datasets with reasonable number of features or classes. Moreover, depending on the set-up, this might be very costly for tabular data classification when considering hardware requirements to run LLMs.
- the proposed approach seems restrained to applications on which LLMs may have relevant knowledge, hence not too specialized or novel. While this out of the scope of this work, it might be interesting to investigate whether humans may generate prototypes as relevant as the LLMs on some datasets.
- The proposed approach is restrained to classification and cannot be directly adapted for regression tasks.

**Questions:**

## Typos:
Authors might consider going over their manuscript to correct typos which occur quite frequently. In particular, citations should be placed after a word and separated with by a whitespace (e.g. line 346 "*STUNT(Nam et al., 2023),*" -> "*STUNT (Nam et al., 2023),*").


## Questions
- **(i)** The $\tau$ parameter in equation $3$ appears critical. **How is $\tau$ selected ?**
- **(ii) Have the authors investigated the impact of the number of paramaters of the base LLM on the overall performance?** While one would (of course) expect the performance to drop as the the number of parameters of the LLM decreases, it might be interesting to analyse the relationship between performance and number of parameters. In particular, if the performance drop is small between large models and smaller models (e.g. 13B parameter models), this might prove to be a significant argument in favor of ProtoLLM.
- **(iii)** As mentionned as a weaknesses of the proposed method, generating the "*golden prototypes*" requires several iterations that may cause a significant "*training*" time. The authors should consider comparing the overall time required to train the competing models vs the time required to construct the golden prototypes used in inference.
- **(iv)** In figure 5 as the reported results are averaged for different seeds, I would be curious to see the figures including the standard deviations/confidence intervals to assess statistical significance of performance gains. **Have the authors performed any statistcal tests (e.g. Wilcoxon signed-rank test) to compare performance with the other best performing alternative?**
- **(v)** As mentionned as a weakness of the proposed approach, ProtoLLM cannot directly be applied for regression. **Have the authors considered modifications/adaptation to the proposed setup to possibly perform regression tasks?** If so, could they share their opinion on the faisability?
- **(vi)** In [1], the authors analyze the impact of spurious correlation on the performance of their proposed method in comparison to existing methods. While ProtoLLM works for zero-shot classification, **have the authors analyzed the effect of including those noisy samples when generating the golden prototypes in the few-shot setting?**
- **(vii)** I might be interesting to analyze how ProtoLLM performs on datasets that have a high number of features, e.g. Arrhythmia (https://archive.ics.uci.edu/dataset/5/arrhythmia) or other tabular datasets the authors might think of.
- **(viii)** For the few-shot settings, to assess the added value of the proposed method, one may consider the following setting: For each class, construct the *prototype* using **only** the samples and excluding the prototype generated by the LLM. That way one could assess the explicit added value of the prototype in the few-shot setting as done for the zero-shot setting.

Given the listed strength and weaknesses, as well as questions and interrogations listed above, we lean towards reject. However, we are open to discussion and would happily increase our score if the authors address our concerns and interrogations.

[1] Sungwon Han, Jinsung Yoon, Sercan O Arik, and Tomas Pfister Large language models can automatically engineer features for few-shot tabular learning. In Forty-first International Conference on Machine Learning, 2024.

---

> ### Author Response · Authors · 2024-11-25
> **Thank you for your detailed and valuable review(1/3)**
>
> We appreciate the reviewer’s feedback and have provided the following responses to address the concerns raised about our paper. Please also check out the revised version of our manuscript (we have highlighted the revisions in blue), where notations, typos, and statements have been refined and revised for better understanding.
>
> **While the proposed approach is novel and interesting, the contribution is relatively thin for a conference like ICLR**
> >Firstly, we appreciate the reviewer's positive comment about our novelty. We hold the belief that our ProtoLLM in this paper gives an interesting insight for introducing LLMs into the zero and few-shot tabular data analysis. We highlight our contributions as follows:
> >- Few-shot tabular learning has long been a popular research topic, due to its wide variety of applications. However, the features in tabular data are often heterogeneous, and their inter-relationships are not inherently sequential. This complexity poses significant challenges for tabular data learning, particularly in real-world few-shot constraints,
> That is to say, proposing an effective method for few-shot tabular learning is necessary but non-trivial.
> >- Recently, the impressive performance of LLMs has highlighted their broad knowledge and potential in instruction following and low-shot understanding. This motivated recent attempts to integrate LLMs with tabular data
> learning and our method belongs to this group, which is a novel and interesting solution and usually outperforms conventional tabular learning baselines.
> >- However, existing LLM-based tabular few-shot methods usually prompt the LLM with the few-shot samples. Our work observed an unusual phenomenon: directly using LLMs for data augmentation or rule generation by feeding a few examples significantly degrades the reasoning ability in tabular data understanding.
> >- To this end, we propose an example-free framework to leverage the inherent knowledge of LLMs. Our key idea is to prompt the LLM for feature value generation based solely on task and feature description. Without such example pollution, each output feature value is treated as a standard guideline, and they together act as a prototype for each class. To transfer the LLM’s knowledge to a given task, we further design an efficient fusion strategy to integrate the prototype with examples, which requires no learnable variable, i.e. neither fine-tuning LLM nor learning a classification model.
> >- To sum up, querying LLM with our designed prompt is zero-shot, and applying ours to tabular data classification is training-free. It is non-trivial since most of existing methods usually need to depends on the training samples or even finetune the LLM. Therefore, ours is a novel example-free and training-free framework for tabular data learning, which will provide a new way about how to integrate the LLMs into tabular data learning.
>
> **Computing cost**
> >Thank you for your comments. We would like to clarify that our approach requires only $d \times K$ queries, where $d$ is the number of features and $K$ is the number of iterations. That is, each query prompts the LLM to generate feature values for all categories simultaneously. Moreover, when using an LLM like GPT-3.5, there are no hardware requirements since the queries are handled through the corresponding API. Furthermore, as shown in Table 1, we report the average number of tokens per query. We find that our token consumption is low and efficient even when using local LLMs. For datasets with many features, previous work, such as TabLLM and FeatLLM, requires all features to be fed into the LLM, which may exceed the model's limitations on input length. Our ProtoLLM can query the LLM feature by feature, providing an alternative for such datasets.
>
> ### Table 1: Average number of tokens per query across datasets
>
> | Dataset | Adult| Bank | Blood | Car |
> |--|--|--|---|---|
> | **ProtoLLM** | 534.7| 454.4 |482.6 | 572.2 |
> | **FeatLLM-rule** | 1254.7| 903.1 | 765.9 | 610.7 |
> | **FeatLLM-function** | 3018.7 | 2637.6 | 1320.6 |1476.6 |
>
> **Novel feature generation**
> >We first want to note that employing LLMs for few-shot classification is gaining more and more research attention recently[1-2]. Pre-trained on web-scale tokens and a huge number of GPUs, LLMs show human-like abilities for understanding and reasoning. We in this paper assume that LLMs have relevant knowledge in most daily applications. Given the fact that Employing humans to generate the prototypes can be more costly in terms of time and money, which is out of the scope of this paper. Fortunately, recent research [3] finds that LLMs outperform human experts in most basic tasks.
>
> [1] Large Language Models Can Automatically Engineer Features for Few-Shot Tabular Learning.
>
> [2] TabLLM: Few-shot Classification of Tabular Data with Large Language Models.
>
> [3] Jones, Nicola. "AI now beats humans at basic tasks—new benchmarks are needed, says major report." Nature 628.8009 (2024): 700-701.

---

> ### Author Response · Authors · 2024-11-25
> **Thank you for your detailed and valuable review(2/3)**
>
> **Results on regression tasks**
> >We agree with you that the tabular regression task is a significant application area for tabular learning. One of the possible directions to apply our ProtoLLM to regression tasks is viewing our generated features as a data augmentation tool. For example, we can concatenate all generated values feature-by-feature to form a virtual labeled sample. Together with few-shot labeled samples, we can improve the regression performance of baselines. We here report the root mean squared error (RMSE) score on abalone and  California datasets in Table 2. We find that our ProtoLLM improves the baseline by a large margin in most cases. This demonstrates the potential of ProtoLLM for regression tasks. Note that it is a simple attempt to apply our ProtoLLM to regression tasks, we leave it as future work for more systematic and in-depth analysis.
>
> ### Table 2: RMSE results on regression task (lower is better)
>
> | Dataset   | Shot | KnnRegressor (w/o ProtoLLM) | KnnRegressor (w/ ProtoLLM) | LinearRegressor (w/o ProtoLLM) | LinearRegressor (w/ ProtoLLM) | MLP (w/o ProtoLLM)  | MLP (w/ ProtoLLM)    |
> |---|--|----|------|---|----|---|---|
> | **Abalone**   | 4    | 3.68    | **3.59**    | 4.01    | **3.63**    | 8.15   | **3.25**    |
> |     | 8    | **3.40**   | **3.40**   | 8.24   | **3.31**  | 4.84  | **3.00**   |
> |    | 16   | 3.30  | **3.13** | 3.67   | **3.03**    | 3.68 | **2.80** |
> | **California**| 4    | 127695.29  | **124478.21** | 145197.65    | **121296.77**    | 175730.68     | **168763.81**  |
> |    | 8    | 122038.60   | **117945.30**   | 164740.07   | **115216.90**    | **165226.72** | 168103.52  |
> |      | 16   | 119206.11   | **115849.27**  | 156167.49  | **111100.62**  | **164411.93** | 169072.75      |
>
> **The selection of $\tau$**
> > We did not explicitly select the $\tau$ parameter. Instead, we just set it to 1 as a fixed value in our experiments.
>
> **Impact of the number of parameters of the base LLM**
> >Thank you for your insightful question. We have investigated the impact of the number of parameters of the base LLM on overall performance, where we consider Llama-3B, Llama-8B models, GPT-3.5 and GPT-4O. The detailed results can be found in Table 3. In general, the performance of Llama-8B is better than that of Llama-3B, but the difference is not substantial. Moreover, we find that Llama-3B still achieves good results, indicating that even smaller models can perform well with ProtoLLM.
>
> ### Table 3: Comparison of AUC scores with different LLMs
>
> | Dataset    | Shot | Llama3B | Llama8B | GPT-3.5 | GPT-4o |
> |------------|------|---------|---------|---------|--------|
> | **Adult**  | 0    | 80.85   | 84.78   | 85.93   | **85.74** |
> |            | 4    | 81.87   | 84.87   | **86.01** | 85.82   |
> |            | 8    | 82.76   | 85.05   | **86.12** | 85.91   |
> |            | 16   | 83.59   | 85.10   | **86.28** | 85.94   |
> | **Blood**  | 0    | 73.34   | **77.91** | 75.63   | 77.83   |
> |            | 4    | 75.16   | 77.15   | 75.98   | **77.17** |
> |            | 8    | 76.03   | **76.99** | 76.35   | **76.99** |
> |            | 16   | **76.69** | 76.01   | 75.46   | 76.21   |
> | **Diabetes** | 0  | **79.84** | 72.67   | 75.65   | 73.17   |
> |            | 4    | **80.02** | 73.06   | 75.68   | 73.85   |
> |            | 8    | **79.95** | 73.67   | 75.76   | 74.16   |
> |            | 16   | **79.91** | 73.60   | 75.70   | 74.08   |
> | **Heart**  | 0    | 63.30   | 64.33   | 58.28   | **74.37** |
> |            | 4    | 70.79   | 73.03   | 65.40   | **79.30** |
> |            | 8    | 75.51   | 77.95   | 70.55   | **82.09** |
> |            | 16   | 82.04   | 84.13   | 78.09   | **85.36** |
>
> **Training and test time**
> >Following your advice, we report the training and testing time of different methods in Table 4. We find that our ProtoLLM requires higher training time costs. However, our ProtoLLM shows faster inference time than others (especially Tablet and Tabllm) thanks to the efficient prototype-based classification strategy we designed.
>
> ### Table 4: Training and Inference Time of Different Methods
>
> | **Model**   | **Training (in seconds)** | **Inference (in milliseconds)** |
> |------|----|------|
> | LogReg   | 0.721   | 0.001     |
> | XGBoost          | 28.512   | 0.006      |
> | RandomForest     | 1.343      | 0.001         |
> | SCARF            | 426.859      | 0.002        |
> | TabPFN           | 0.44                     | 1.149                          |
> | STUNT            | 642.796                  | 0.006                          |
> | In-context†      | N/A        | 463         |
> | TABLET           | 0.813                    | 523.254                        |
> | TabLLM           | 251.242                  | 335.127                        |
> | FeatLLM          | 860.094                  | 0.006                          |
> | **ProtoLLM**     | **1364.38**              | **0.001**                      |

---

> ### Author Response · Authors · 2024-11-25
> **Thank you for your detailed and valuable review(1/3)**
>
> **Results of Fig.5**
> >We kindly remind you that the standard deviations and the full results of Figure 5 were reported in Table 9 of the appendix. Additionally, following your suggestion, we have performed a Wilcoxon signed-rank test to compare the performance of our method with the other baselines. The results show statistically significant differences with $p < 0.05$, corresponding to a 95\% confidence level. The details of this statistical analysis are provided in Table 5.
>
> ### Table 5: Wilcoxon Signed-Rank Test
>
> | **Model**         | **p**         |
> |--------------------|---------------|
> | **LogReg**        | 1.45e-06      |
> | **XGBoost**       | 5.61e-11      |
> | **KNN**           | 6.62e-12      |
> | **MLP**           | 5.32e-10      |
> | **TabPFN**        | 6.11e-06      |
> | **STUNT**         | 3.73e-07      |
> | **In-context**    | 1.02e-09      |
> | **TABLET**        | 1.32e-08      |
> | **TabLLM**        | 8.81e-03      |
> | **FeatLLM**       | 1.15e-02      |
>
> **Results on noisy samples**
> >Following your suggestion, we analyze the impact of spurious correlations. Specifically, we conduct experiments by randomly selecting columns from the Adult dataset and applying them to the Bank dataset. The Adult dataset predicts a person’s income and its features may introduce spurious correlations that could affect the task in Bank, which involves predicting the likelihood of a person subscribing to a term deposit. For this analysis, we set the number of shots to 4 and included 0-5 columns from the Adult dataset. The performance changes are shown in Table 6. As the number of spurious columns increases, the performance of ProtoLLM decreases gradually. This is because the spurious columns negatively impact the model's performance. However, as proposed in Section 4.3 of our paper, by assigning weights queried from LLMs to each feature (i.e., ProtoLLM-weighted), we can mitigate this effect and reduce the influence of spurious correlations.
>
> ### Table 6: The Impact of Spurious Correlations on AUC Results
>
> | Number of Spurious Columns | LogReg | KNN | TabPFN| FeatLLM | ProtoLLM | ProtoLLM-Weighted |
> |--------------------------------|------------|----------|------------|-------------|--------------|------------------------|
> | **0**                          | 65.63      | 63.13    | 69.07      | 74.16       | 81.41        | 81.06                 |
> | **1**                          | 65.21      | 63.07    | 68.79      | 74.43       | 80.57        | 82.28                 |
> | **2**                          | 64.28      | 62.84    | 67.83      | 73.95       | 79.89        | 81.87                 |
> | **3**                          | 64.60      | 62.13    | 67.44      | 71.60       | 79.39        | 81.66                 |
> | **4**                          | 63.62      | 62.04    | 66.98      | 73.02       | 78.69        | 81.50                 |
> | **5**                          | 63.49      | 61.81    | 66.14      | 71.57       | 78.28        | 81.76                 |
>
> **Results on Arrhythmia dataset**
> >Following your advice, we reported results on Arrhythmia, as listed in Table. 7, We find that our models are still available on such datasets with many features and outperform baselines. Unlike previous works such as TabLLM which need to feed all features into LLMs, which is restricted by the input length, our approach queries LLMs feature-by-feature and thus can work on datasets with a large number of features.
>
> ### Table 7:  Results of Different Methods on Arrhythmia dataset
>
> *Note: N/A indicates that the method cannot process this dataset.*
>
> | Dataset      | Shots | LogReg   | KNN   | TabPFN | FeatLLM | ProtoLLM  |
> |-----|---------|------|-----|---|---|--|
> | Arrhythmia    | 1    | 72.16        | 52.48         | N/A        | N/A         | **72.49**      |
> |                   | 2   | 77.70        | 67.01         | N/A        | N/A         | **78.11**      |
>
> **LLM-based prototypes**
> >Thank you for your suggestion. In Figure 6 of our paper, we illustrated the relationship between performance (y-axis) and the number of queries (x-axis). $K = 0$ in the X-axis means that we only use the training samples to construct the prototype, exactly the setting described by the reviewer. We can find that, for different few-shot settings, the performance of $K = 0$ is lower than that of $K > 0$, showing the effectiveness of the prototype generated by the LLM.

---

### Official Review · Reviewer_8Ebi · 2024-10-28

**Soundness:** 2
**Presentation:** 3
**Contribution:** 2
**Rating:** 5
**Confidence:** 4

**Summary:**

Conventional machine learning models for tabular data rely on labeled samples, and performance tends to degrade when few samples are available. It has recently been argued that LLMs may learn useful information for certain tabular classification tasks during pretraining. This may make them superior classifiers in the extreme few-shot setting (which, for tabular data, could roughly be classified as fewer than 100 labeled datapoints available for training). TabLLM and FeatLLM constitute two important precursor works in this line of research.

The authors of ProtoLLM remove the few-shot examples from the prompt template and prompt LLMs to generate oracle feature values according to the LLM's inherent knowledge about the given tasks. They query LLMs feature by feature, rather than generating all features of a tabular sample simultaneously. Instead of generating a single value for each feature, they query LLMs K = 10 times for each feature across all datasets. Results are provided for zero-shot and few-shot.

The authors claim their method leads to robust and superior performance on a range of datasets.

**Strengths:**

* The authors provide some intriguing indications that LLMs can solve certain tabular classification problems which are well specified using natural language zero-shot, simply by predicting likely feature values for each class.
* The authors include some helpful ablation studies and appendix material, including a complete results table.
* The choice of few-shot baseline methods is comprehensive and strong.

**Weaknesses:**

MAJOR

* This paper argues for the scientific relevance of a method whose main results are demonstrated only on ChatGPT, a closed-source LLM. Closed models make bad baselines, are not reproducible, and bias future research [https://hackingsemantics.xyz/2023/closed-baselines/]. To make the claim that ProtoLLM is significant, it is necessary to show that it is doing something qualitatively different which improves results while controlling for model size, the type and amount of data, and important hyperparameters. Since we don’t have any of this information for GPT, it is not possible to claim ProtoLLM improves on prior work; apparent improvements could simply reflect a quirk of the chosen GPT checkpoint, for example. At a bare minimum, the authors should show results on a small open-source LLM such as T0 or LLAMA-3-8B in addition to GPT.
* The choice of datasets is small and non-standard; using a large established benchmark in the literature would lend more credibility to the findings [https://arxiv.org/abs/2305.02997]. Data contamination is a major concern here, as most of the datasets included are old. On the two datasets where data contamination is less likely (NHANES and Cultivars), a simple logistic regression baseline with just 64 data points beats ProtoLLM.
* Some datasets in Fig. 5 have missing baselines.
* Something is wrong with the formatting of this paper; line numbers don't align with the lines.
* There are few zero-shot baselines provided; DSPy would have been a natural choice [https://github.com/stanfordnlp/dspy]. The one zero-shot baselines that is provided (TabLLM GPT-3.5) beats ProtoLLM on 4 of 9 datasets; given that the authors argue their contributions are most significant in the zero-shot regime, this is troubling.
* TabLLM GPT-3.5 results should be made available for few shot as well as zero-shot for a fair comparison between methods.

MINOR

* Fig. 5 is hard to read; it should be larger.
* Please provide links to the datasets used in the appendix; there are many versions of these datasets in common use.
* Many typos (lines 283, 300, Table 1, Sec. 4.1 header, line 337, etc).

**Questions:**

* How does the method deal with continuous features? Is it similar to FeatLLM?
* How many tokens are consumed for an average iteration of ProtoLLM?
* How is the distance function used in ProtoLLM? What is meant by 'logistics' in this sentence: "All we need to calculate the logistics is the Euclidean distance between the input samples and obtained prototype"?

---

> ### Author Response · Authors · 2024-11-25
> **Thank you for your detailed and valuable review(1/2)**
>
> We appreciate the reviewer’s feedback and have provided the following responses to address the concerns raised about our paper. Please also check out the revised version of our manuscript (we have highlighted the revisions in blue), where notations, typos, and statements have been refined and revised for better understanding.
>
> **Additional results on open-source LLMs**
> >Thank you for your valuable feedback. We understand the concerns surrounding the use of closed-source models like ChatGPT. However, we would like to highlight that a range of recent studies have successfully applied closed-source large models for few-shot classification [1-3] and other tasks such as feature engineering [4-5]. In response to your suggestion, we have expanded our experiments to include open-source LLMs, specifically LLaMA-3B and LLaMA-8B, to further validate our approach. However, as noted by Hugging Face, the T0* models are not well-suited for code-related tasks, which is why they were not included. More details on these experiments can be found in Table 1. We find that our ProtoLLM still achieves comparable results on open-source LLMs, showing the robustness of different base LLMs.
>
> ### Table 1: Comparison of AUC scores with different LLMs
>
> | Dataset    | Shot | Llama3B | Llama8B | GPT-3.5 | GPT-4o |
> |------------|------|---------|---------|---------|--------|
> | **Adult**  | 0    | 80.85   | 84.78   | 85.93   | **85.74** |
> |            | 4    | 81.87   | 84.87   | **86.01** | 85.82   |
> |            | 8    | 82.76   | 85.05   | **86.12** | 85.91   |
> |            | 16   | 83.59   | 85.10   | **86.28** | 85.94   |
> | **Blood**  | 0    | 73.34   | **77.91** | 75.63   | 77.83   |
> |            | 4    | 75.16   | 77.15   | 75.98   | **77.17** |
> |            | 8    | 76.03   | **76.99** | 76.35   | **76.99** |
> |            | 16   | **76.69** | 76.01   | 75.46   | 76.21   |
> | **Diabetes** | 0  | **79.84** | 72.67   | 75.65   | 73.17   |
> |            | 4    | **80.02** | 73.06   | 75.68   | 73.85   |
> |            | 8    | **79.95** | 73.67   | 75.76   | 74.16   |
> |            | 16   | **79.91** | 73.60   | 75.70   | 74.08   |
> | **Heart**  | 0    | 63.30   | 64.33   | 58.28   | **74.37** |
> |            | 4    | 70.79   | 73.03   | 65.40   | **79.30** |
> |            | 8    | 75.51   | 77.95   | 70.55   | **82.09** |
> |            | 16   | 82.04   | 84.13   | 78.09   | **85.36** |
>
> **The datasets are small and non-standard**
> >Our method is specifically designed for few-shot classification tasks, and we follow previous work[3] for dataset selection. Besides, the datasets used in ProtoLLM have also been used in other prior studies[2,6]. We also would like to highlight that our ProtoLLM beats LogReg and other LLM-based methods in most cases on NHANES and Cultivars datasets on both zero-shot (learning-based algorithms, such as LogReg fails to work) and very few-shot settings (len then 32). This demonstrates the effectiveness of our approach in zero or few-shot tasks, even on novel datasets.
> >
> >Following your suggestion, we report the results on Arrhythmia, Soybean, and Cmc from TabZilla Benchmark in Table 2. We find that our ProtoLLM still achieves the best results in most cases when compared to traditional and LLM-based models, which increases the credibility of the proposed method.
>
> ### Table 2: AUC score of different models on TabZilla Benchmark, where N/A means the method fails to run on such datasets due to the many examples or classes.
>
> | Dataset  | Shots | **LogReg** | **KNN** | **TabPFN** | **FeatLLM** | **ProtoLLM** |
> |--|---|---|--|----|--|---|
> | **Arrhythmia**| 1   | 72.16  | 52.48   | N/A | N/A  | **72.49** |
> |  | 2   | 77.70      | 67.01   | N/A  | N/A  | **78.11**    |
> | **Soybean**   | 1  | 94.31  | 52.13 | N/A  | N/A | **94.44**  |
> | | 2   | 96.29 | 52.04   | N/A  | N/A  | **97.20**  |
> | **Cmc**  | 1 | 52.48 | 52.13   | 54.13  | 53.88 | **54.39**  |
> |  | 2  | 54.14  | 52.04   | 53.84 | **56.37** | 54.72  |
>
>
> [1] Language models are weak learners, NeurIPS 2023
>
> [2] Tabular Transfer Learning via Prompting LLMs, COLM 2024
>
> [3] Large Language Models Can Automatically Engineer Features for Few-Shot Tabular Learning, ICML 2024
>
> [4] Large Language Models for Automated Data Science: Introducing CAAFE for Context-Aware Automated Feature Engineering, NeurIPS 2023
>
> [5] Optimized Feature Generation for Tabular Data via LLMswith Decision Tree Reasoning, NeurIPS 2024
>
> [6] TabLLM: Few-shot Classification of Tabular Data with Large Language Models, AISTATS 2023

---

> ### Author Response · Authors · 2024-11-25
> **Thank you for your detailed and valuable review(2/2)**
>
> **The missing baselines in Fig.5**
> >Thank you for your observation. The absence of certain baselines in Fig. 5 is primarily due to the limitations of some baseline frameworks. For example, many LLM-based frameworks, e.g., In-context, TABLET, and TabLLM, have restrictions on the maximum number of input tokens they can handle. This limitation makes it challenging or even infeasible to include these baselines in scenarios with a higher number of shots or more complex features. Additionally, TabPFN cannot run on datasets with a large number of features. Instead, ours has the ability to handle scenarios with a higher number of shots or more complex features because our designed example-free prompt is efficient. We appreciate you pointing this out and will clarify this in the revised manuscript to avoid such confusion.
>
> **More comparison on zero-shot tasks**
> >Thank you for your valuable feedback. We appreciate your suggestion regarding zero-shot baselines, and we here incorporate additional zero-shot experiments in Table 3. Specifically, DSPy and P2T have been included as baselines. We find that our ProtoLLM outperforms DSPy and P2T in all cases, which shows the efficacy of the proposed model on zero-shot tabular data analysis.
>
> ### Table 3: Zero-shot results with more baselines. GPT-3.5 is used for all the baselines.
>
> | Dataset    | DSPy  | P2T   | ProtoLLM(ours) |
> |----|-------|---|----|
> | **Bank**   | 53.14  | 73.08 | **80.20**   |
> | **Blood**  | 50.33  | 68.64 | **75.63**    |
> | **Credit-g** | 45.60  | 55.28 | **61.29**  |
> | **Diabetes** | 64.87  | 72.55 | **75.65**   |
>
> **TabLLM (GPT-3.5) VS ProtoLLM on zero-shot settings**
> > We list the numerical results of ProtoLLM and TabLLM(GPT-3.5) in Table 4 for clear comparison. We find that **1)** Although ours is inferior to TabLLM (GPT3.5) on 4 of 9 datasets, the performance of ProtoLLM is still comparable to TabLLM (GPT3.5), especially on Adult and Diabetes; **2)** Our ProtoLLM beats TabLLM by a large margin on 5/9 datasets. These findings prove the effectiveness of ours in the zero-shot setting.
>
> ### Table 4: Comparison of AUC scores between ProtoLLM and TabLLM (GPT-3.5), including improvement of ProtoLLM.
>
> |    | Adult  | Bank  | Blood | Car   | Credit-g | Cultivars | Diabetes | Heart  | NHANES |
> |---|---|---|---|----|---|----|-----|----|---|
> | **TabLLM(GPT-3.5)** | 87.24  | 69.54 | 61.51 | 75.80 | 45.88    | 49.30  | 79.92    | 64.62  | 94.53  |
> | **ProtoLLM** | 85.93  | 80.20 | 75.63 | 78.29 | 61.29  | 58.93  | 75.65| 58.28  | 83.96  |
> | **Improvement** | -1.31  | 10.66 | 14.12 | 2.49  | 15.41  | 9.63  | -4.27    | -6.34   | -10.57 |
>
> **TabLLM GPT-3.5 results for few-shot settings**
> >TabLLM is proposed to solve tabular few-shot classification tasks, which requires fine-tuning LLM with few-shot samples and utilizes T0 as the LLM backbone.  For the zero-shot setting, as TabLLM is also feasible, we report the zero-shot performance of TabLLM with GPT-3.5 for a fair comparison. For the few-shot setting, TabLLM involves the fine-tuning of LLM but GPT-3.5 is a  closed-source LLM, making it irrealizable to provide few-shot results for GPT-3.5. However, we kindly remind you that, in Fig. 5 in the manuscript, we reported the few-shot classification results of TabLLM with T0 being the backbone.
>
> **Minor issues**
> >**1)** Thank you for pointing this out. We have revised Fig. 5  to make it clearer to read; **2)** The links to each dataset are added in the revision; **3)** Other typos and notations are corrected in the revision, please check out our revised version for details.
>
> **Continuous features**
> >Unlike FeatLLM which directly uses the numerical values,  we add the z-score normalization for the generated continuous features to complete the prototypes.
>
> **How many tokens are consumed**
> > Our token consumption is demonstrated in Table 5. We compare our method with FeatLLM, which uses LLMs to extract rules (FeatLLM-rule) and functions (FeatLLM-function), respectively. Due to the designed example-free prompt, ProtoLLM achieves significantly lower token usage compared to both FeatLLM, highlighting its efficiency.
>
> ### Table 5: Average number of tokens per query across datasets
>
> | Dataset   | **Adult** | **Bank** | **Blood** | **Car** |
> |--|---|----|---|---|
> | **ProtoLLM**  | 534.7 | 454.4 | 482.6     | 572.2   |
> | **FeatLLM-rule** | 1254.7| 903.1    | 765.9     | 610.7   |
> | **FeatLLM-function** | 3018.7 | 2637.6   | 1320.6    | 1476.6  |
>
> **Distance function used in ProtoLLM**
> >ProtoLLM specifies the distance function as Euclidean distance by default. We also performed ablation experiments on Manhattan and cosine distance. The results in Table 2 of the submission show the robustness of our ProtoLLM to different distance functions.
>
> **Logistics at line 99**
> >We have replaced "logistics" with "prediction probability" in the revision: ``All we need to calculate the prediction probability is the Euclidean distance between the input samples and obtained prototypes.``

---

> > ### Comment · Reviewer_8Ebi · 2024-11-26
> >
> > SUMMARY: The authors have presented a strong rebuttal and added useful content to their paper which has addressed some of my concerns. Therefore, I have raised my score from a 3 to a 5.
> >
> > REVIEW OF WEAKNESSES:
> >
> > W1. Addressed, thank you.
> >
> > W2. Not yet addressed. The provided experiments do not address my comment, which was that `using a large established benchmark in the literature would lend more credibility to the findings`. Using a benchmark means using it in its entirety, not cherry-picking datasets. This would be like selecting three classes from ImageNet on which your model does well and claiming you had reported results for ImageNet. And if you're going to cherry pick, why would you pick datasets on which your chosen baselines fail to run?
> >
> > As for the matter of contamination, your Fig. 5(k) shows LogReg (and other baselines) outperforming ProtoLLM on NHANES after just 16 samples, as far as I can tell. But whether it's 16 or 32, that many labeled samples is a pretty low bar to clear for tabular data, and it's the trend line of declining relative performance for your method that is troubling, as it suggests a scaling limitation. I think that a Limitations section in the main paper and a few discussion points would make your paper more persuasive, because it does appear to perform strongly in the zero and extreme few-shot regimes.
> >
> > W3. Addressed, thank you.
> >
> > W4. Not yet addressed. The line numbers are still misaligned in the revised manuscript.
> >
> > W5. Partially addressed. I don't see these updates in the revised manuscript, please confirm that they will be added by the camera ready deadline. Also, please add the missing datasets by the camera ready deadline. It would also be helpful, again, to have a Limitations section with some analysis on why your method sometimes underperforms and sometimes overperforms, to inform future work.
> >
> > W6. Not yet addressed. The Fig. 5 comparison between few-shot TabLLM and ProtoLLM disadvantages the former unfairly by comparing a T0 LLM backbone to a GPT LLM backbone, which is like pitting a squirt gun against a water cannon. You had the opportunity both when writing the paper and during the rebuttal period to demonstrate that ProtoLLM could outperform few-shot TabLLM when using the same T0 backbone, but you chose not to run these experiments.

---

> ### Author Response · Authors · 2024-11-27
> **Response to Reviewer 8Ebi:**
>
> Thank you very much for your positive feedback and detailed comments on our response.  Your valuable response helped us identify the limitations of our response at the first stage. This gives us an opportunity to improve the quality of our submission once again. We are grateful for the time and effort you have invested in reviewing our work.
>
> **Results on tabZilla benchmark**
> >We apologize for misinterpreting your suggestion during the first rebuttal stage. We only reported results on three datasets rather than the whole tabZilla benchmark. We hope the following explanation will address your contents:
> >- First, we kindly remind you that we did not cherry-pick datasets at the first rebuttal stage. We experimented on the Soybean, Arrhythmia, and Cmc datasets, mainly because these datasets fulfilled the conditions of multi-class and multi-feature, among others, and were in line with the recommendations of the other reviewers.
> >- Most LLM-based models, such as FeatLLM and our ProtoLLM, require feature descriptions to improve the LLMs' understanding. We find 18 datasets within the tabZilla benchmark that have this type of information attached to them. We are in the process of comparing more of these datasets and will report the results as soon as possible (in 4 days).
>
> **Response to limitations**
> >Thanks for your suggestion. We have added the discussions about the limitations in Appendix A.13 due to the limited space and we will add the section in the main paper in the camera ready. Below, we provide the details about limitations for your convenience: This work introduces an example-free and training-free  ProtoLLM for zero and few-shot tabular classification task. It prompts the LLM for feature value generation based solely on task and feature description, making it difficult to apply to datasets without such a describing information. Besides, our ProtoLLM prompts the LLM without the training examples and designs the training-free prototype for classification instead of training a classifier. Therefore, it is more suitable for the zero and extreme few-shot regimes. With the increasing number of training samples, our approach is gradually closer to or inferior to other methods that depends on training samples and a learnable classifier, which is an interesting direction and can be considered in future work.
>
>
> **Response to comparison with TabLLM**
> >For T0 models, they are not well-suited for code-related tasks {https://huggingface.co/bigscience/T0}. To achieve automatically feature value extraction and make the whole algorithm end-to-end, we ask LLMs to generate structure answers with the markdown code snippet formatted, including the leading and trailing "```json" and "```", as shown in Figure 3 in our submitted version. Therefore, using T0 as the backbone is not suitable for ours. We agree with you that GPT-3.5 is more powerful than T0. To address your concerns further, we consider conducting experiments on a relatively fair backbone, such as Llama3B. It is in the process and we will report the results as soon as possible (in 4 days).
>
> **Response to others**
> >We have added the results of zero-shot baselines in Appendix A.5 in revision. Besides, we will add the missing datasets by the camera ready deadline.

---

> ### Author Response · Authors · 2024-12-01
> **Response to Reviewer 8Ebi:**
>
> **Response to Benchmark**
>
> Following your suggestion, we further conducted experiments on the remaining datasets from tabZilla benchmark, where we consider datasets that contain the necessary descriptive information, such as feature names and descriptions. We report the results at Table 1 (Due to the complex table structure, please refer to the anonymous page for detailed results{https://anonymous.4open.science/r/anonymous-2D04/rebuttle_results.pdf}.
>
> **Table 1: Additional datasets on tabZilla benchmark
> | Datasets   | Numerical | Categorical | Objects | Classes | Shot | LogReg         | KNN            | TabPFN         | FeatLLM        | ProtoLLM        |
> |------------|-----------|-------------|---------|---------|------|----------------|----------------|----------------|----------------|-----------------|
> | **arrhythmia** | 206       | 73          | 452     | 13      | 1    | _72.16±9.09_   | 52.48±0.00     | N/A            | N/A            | **72.49±5.24**  |
> |            |           |             |         |         | 2    | _77.70±3.11_   | 67.01±2.19     | N/A            | N/A            | **78.11±4.33**  |
> | **ecoli**     | 7         | 0           | 336     | 8       | 1    | _91.68±5.41_   | 70.13±0.00     | 86.65±7.32     | 58.87±0.00     | **93.22±3.45**  |
> |            |           |             |         |         | 2    | _92.94±4.76_   | 89.92±6.30     | 92.00±4.80     | N/A            | **95.93±2.79**  |
> | **lymph**     | 3         | 15          | 148     | 4       | 1    | **86.13±5.74** | 81.36±3.99     | 84.61±7.29     | 85.71±8.53     | _85.98±7.73_    |
> |            |           |             |         |         | 2    | _87.64±8.05_   | 83.74±7.79     | 83.08±10.21    | 87.31±7.33     | **90.78±6.09**  |
> |...|
> |...|
> | **average rank**|           |             |         |         | 1    | 2.85           | 4.69           | 3.31           | 2.54           | **1.62**        |
> |                |           |             |         |         | 2    | 3.17           | 4.58           | 3.17           | 2.17           | **1.92**        |
> |                |           |             |         |         | 4    | 3.50           | 5.00           | 2.40           | 2.10           | **2.00**        |
>
> *Note*: The bold text denotes the best performance, while the _underlined_ text indicates the second-best performance. `N/A` indicates that the model has limitations preventing it from running on the dataset.
>
> To summarize, from the results, we find that 1) Overall, Our ProtoLLM beats traditional and LLM-based models in most cases, achieving rankings of 1.62, 1.92, and 2.0, respectively; 2) Because of the efficiency of the introduced training and example-free strategy, our approach can be easily applied in most datasets (Other LLM-based models sometimes fail to run at complex datasets). This demonstrates the flexibility of our ProtoLLM.

---

> ### Author Response · Authors · 2024-12-01
> **Response to Reviewer 8Ebi:**
>
> **Response to comparison with TabLLM**
>
> To address your concerns about a fair comparison with TabLLM, in the second rebuttal phase we found that fine-tuning TabLLM with Llama-3B often leads to poor results, so we decided to report our results for ProtoLLM based on the T0 model, as you had previously suggested.
> To adapt ours to T0, we designed slightly different prompts than those used in our original paper, as T0 is not capable of generating code. The prompt template is as follows:
>
> >Answer choices:{answer choices}
> >
> >{task for a class}. Given a list of {feature name}({feature description}): {feature value list}. Which {feature name} should this belong to
> ?
> >
> >Answer:
>
> Below is an example of a prompt from the Adult dataset, where we query the possible feature values of the relationship
> feature for the class corresponding to individuals earning more than 50,000 dollars per year.
>
> >Answer choices:Own-child|||Husband|||Not-in-family|||Unmarried|||Wife|||Other-relative
> >
> >The person earns more than 50000 dollars per year. Given a list of relationship (what this individual is relative to others):
> Own-child, Husband, Not-in-family, Unmarried, Wife and Other-relative. Which relationship should this belong to?
> >
> >Answer:
>
> For numerical features, we uniformly sample several feature values from the range provided by the few-shot samples to be used
> as answer choices. For categorical features, we list all the categories as answer choices.
>
> Once we obtain the generated feature values from T0 with the designed prompt, we adopt them to build the prototype with Eq.(2) or Eq.(3) and then classify the test samples with Eq.(4) in our submitted version. The results are also provided in {https://anonymous.4open.science/r/anonymous-2D04/rebuttle\_results.pdf}. As shown in table 2, when taking T0 as the backbone like TabLLM, our proposed method with the degraded prompt performs better than TabLLM. It proves the effectiveness and robustness of ours, which avoids fine-tuning LLM and is available for different backbones. We will add the results into our camera ready. Thanks for your suggestions, which are helpful for improving the quality of our work.

---

### Official Review · Reviewer_ToLN · 2024-11-02

**Soundness:** 2
**Presentation:** 3
**Contribution:** 2
**Rating:** 6
**Confidence:** 5

**Summary:**

This paper suggests generating individual tabular feature values with LLM (which they denote as Oracle feature generation) per class, then building a prototype, which is used to build a non-parametric classifier. This prototype can be jointly used with few-shot samples by building prototypes with few-shot samples (i.e., averaged feature value of few-shot samples). The author shows that the method is effective for zero to low-shot (e.g., 64-shot) classification for tabular dataset.

**Strengths:**

**Strength**

1. The overall performance is strong.

2. The method may offer novelty from one perspective: generating features with LLMs already exists [1], but using them additionally for building prototypes is relatively new.

**Reference**\
[1] Large Language Models Can Automatically Engineer Features for Few-Shot Tabular Learning, ICML 2024

**Weaknesses:**

**Weakness**

1. The writing lacks clarity and is not mathematically formulated. For example, there are several questions:
- What is the feature value z? This value should be mathematically defined. Currently, it appears to be the feature generated by the LLM. However, it is not clearly defined, making it difficult for readers to understand.
- In Figure 3, 4, can the authors clarify what each color represents?
- Why is the LLM-generated feature referred to as "Oracle"? "Oracle" is a term that requires careful use; I would suggest "LLM-generated feature" instead, as the LLM is not an Oracle.

2. Most of the datasets focus on binary classification, which can be considered a toy example (also, how did the authors compute AUC for the car dataset, which seems to be a multi-class classification task?). The authors should consider including more multi-class classification datasets, such as those in STUNT [1]. Extending this approach to regression tasks is also essential, as it’s a significant application area for tabular learning.

3. The experiment section could be strengthened.
- Variance should be reported for few-shot (or low-shot) learning experiments, as stability is also a crucial aspect of the work.
- Reporting full-shot accuracy and discussing limitations in this setup would add valuable insights.

4. There is a concern regarding dataset contamination and whether the method will work on truly novel datasets. While it’s acknowledged that the authors have considered two datasets new to the LLM, tabular datasets often contain novel features (e.g., from fields like factory settings or biology). It is unclear if this method would generalize well to such data.

5. There are existing works focused on feature generation that can be combined with other effective methods (e.g., XGBoost) to enhance performance further [2,3,4,5]. Typically, in this context, few-shot learning is not the primary focus, as it is somewhat constrained or limited in application.

6. Lack of comparison with LLM used Tabular learning frameworks [6,7,8].

**Summary**\
The paper has certain strengths; however, currently, the weaknesses are more prominent. In this context, I respectfully request the authors to address these issues during the rebuttal.

**Reference**\
[1] STUNT: Few-shot Tabular Learning with Self-generated Tasks from Unlabeled Tables, ICLR 2023\
[2] Large Language Models for Automated Data Science: Introducing CAAFE for Context-Aware Automated Feature Engineering, NeurIPS 2023\
[3] Generalized and Heuristic-Free Feature Construction for Improved Accuracy, SIAM Data Mining 2010\
[4] Learning a Data-Driven Policy Network for Pre-Training Automated Feature Engineering, ICLR 2023\
[5] OpenFE: Automated Feature Generation with Expert-level Performance, ICML 2023\
[6] LIFT: Language-Interfaced Fine-Tuning for Non-Language Machine Learning Tasks, NeurIPS 2022\
[7] Tabular Transfer Learning via Prompting LLMs, COLM 2024\
[8] Language models are weak learners, NeurIPS 2023

**Questions:**

The questions are in the weakness part.

---

> ### Author Response · Authors · 2024-11-25
> **Thank you for your detailed and valuable review (1/3)**
>
> We appreciate the reviewer’s feedback and have provided the following responses to address the concerns raised about our paper. Please also check out the revised version of our manuscript (we have highlighted the revisions in blue), where notations, typos, and statements have been refined and revised for better understanding.
>
> **Writing issues**
> >We thank the reviewer for the helpful suggestion. We have revised our manuscript, where the feature value generation has been mathematically formulated, the captions in Figures 2-4 have been revised, and the "Oracle feature'' has been replaced with "LLM-generated feature''.
> >
> >**What is the value of z?** For a dataset with $D$ features, our ProtoLLM aims to make use of the prior knowledge behind LLMs and ask LLMs to generate the feature values. Let $z_{c,d}$ denote the generated value for $d$-th feature in class $c$:
> >
> > &nbsp; &nbsp; &nbsp; &nbsp; $z_{c,d} = \text{LLM}(P_{d})[c]$, &nbsp; &nbsp; if the $d$-th feature is a numerical feature,
> >
> > &nbsp; &nbsp; &nbsp; &nbsp; $z_{c,d} = \text{One-shot}(\text{LLM}(P_{d})[c])$, &nbsp; &nbsp; if the $d$-th feature is a categorical feature,
> >
> >where $P_{d}$ denotes the prompt input for $d$-th feature, and $\text{LLM}(\cdot)[c]$ denotes the output values of LLMs for $c$-th class. We directly use the output values of LLMs for numerical features and post-process the categorical features with the $\text{One-shot}(\cdot)$ function to convert the output class index to a one-hot vector.
> >
> >**colors in Figure 3, 4** Figure 3 shows an example of our designed prompt for ProtoLLM, which consists of two parts: <Meta-Info> (red color) and <Query> (green color). the blue-colored sentences denote the response formation. Figure 4 shows an example of the output of our ProtoLLM, where the blue-colored words denote the class names, and the red-colored sentences denote the corresponding output values of LLMs.
>
> **Experiments on multiclass datasets**
> > To test the proposed model on zero and few-shot datasets, we follow previous works[1,2] for dataset selection and metric calculation.
> We find that most previous LLM-based tabular few-shot methods focus on binary classification or multiclass classification with a limited number of classes [1, 3]. Therefore, we initially evaluated our method on these commonly studied datasets.
> >
> >Following your suggestion, we here report additional comparisons on three multiclass datasets from TabZilla Benchmark [5] and STUNT [4]. Table 1 and Table 2 report the statistics of datasets and results of different models respectively. From the results, we would like to note that: **1)** Our ProtoLLM outperforms baselines in most cases, which shows the efficiency of the prototype-based strategy in multiclass settings. **2)** TabPFN often works on small datasets, while FeatLLM inputs all examples into LLM to obtain decision rules, and thus tends to exceed the input length limit on large datasets. As a result, we only report the results of TabPFN and FeatLLM on the Cmc dataset. In opposite, our ProtoLLM queries LLMs feature-by-feature and thus it can process datasets with many features. This shows the flexibility of our approach.
>
> ### Table 1: Statistics of multiclass datasets
>
> |   | **Arrhythmia** | **Soybean** | **Cmc** |
> |---|--|---|----|
> |#Samples| 452 | 683 |1473|
> |#Numerical features| 206   | 0 | 2 |
> |#Categorical features| 73| 35 | 7|
> |#Class| 16 | 19 | 3 |
>
> ### Table 2: Results of different methods on multiclass datasets (mean values with 15 different runs. N/A means the method fails to run on such datasets).
>
> | Dataset  | Shots | **LogReg** | **KNN** | **TabPFN** | **FeatLLM** | **ProtoLLM** |
> |--|---|---|--|----|--|---|
> | **Arrhythmia**| 1   | 72.16  | 52.48   | N/A | N/A  | **72.49** |
> |  | 2   | 77.70      | 67.01   | N/A  | N/A  | **78.11**    |
> | **Soybean**   | 1  | 94.31  | 52.13 | N/A  | N/A | **94.44**  |
> | | 2   | 96.29 | 52.04   | N/A  | N/A  | **97.20**  |
> | **Cmc**  | 1 | 52.48 | 52.13   | 54.13  | 53.88 | **54.39**  |
> |  | 2  | 54.14  | 52.04   | 53.84 | **56.37** | 54.72  |
>
> **AUC calculation on multiclass setting**
> >we follow previous works[1,2] to calculate the AUC score. We find that some datasets are imbalanced. This makes AUC a more appropriate metric for evaluating model performance in these cases, as it better reflects the model's ability to distinguish between classes despite the imbalance.  For multiclass classification, we compute AUC using ``sklearn.metrics.roc\_auc\_score`` with **multi\_class='ovr'**. This computes the AUC of each class against the rest.
>
> [1] TabLLM: Few-shot Classification of Tabular Data with Large Language Models.
>
> [2]  Large Language Models Can Automatically Engineer Features for Few-Shot Tabular Learning.
>
> [3] Tabular Transfer Learning via Prompting LLMs.
>
> [4] STUNT: Few-shot Tabular Learning with Self-generated Tasks from Unlabeled Tables.
>
> [5] When Do Neural Nets Outperform Boosted Trees on Tabular Data?.

---

> ### Author Response · Authors · 2024-11-25
> **Thank you for your detailed and valuable review (2/3)**
>
> **Experiments on regression tasks**
> > We in this paper mainly focus on integrating LLMs into tabular data understanding and following previous works conducted zero and few-shot classification tasks on various tabular datasets. Extensive results demonstrate the improvements and efficiency of the proposed model.
> >
> > We agree with you that the tabular regression task is a significant application area for tabular learning. One of the possible directions to apply our ProtoLLM to regression tasks is viewing our generated features as a data augmentation tool. For example, we can concatenate all generated values feature-by-feature to form a virtual labeled sample. Together with few-shot labeled samples, we can improve the regression performance of baselines. We here report the root mean squared error (RMSE) score on abalone and  California datasets at Table 3. We find that our ProtoLLM improves the baseline by a large margin in most cases. This demonstrates the potential of ProtoLLM for regression tasks. Note that it is a simple attempt to apply our ProtoLLM to regression tasks, we leave it as future work for more systematic and in-depth analysis.
>
> ### Table 3: RMSE results on regression task (lower is better)
>
> | Dataset   | Shot | KnnRegressor (w/o ProtoLLM) | KnnRegressor (w/ ProtoLLM) | LinearRegressor (w/o ProtoLLM) | LinearRegressor (w/ ProtoLLM) | MLP (w/o ProtoLLM)  | MLP (w/ ProtoLLM)    |
> |---|--|----|------|---|----|---|---|
> | **Abalone**   | 4    | 3.68    | **3.59**    | 4.01    | **3.63**    | 8.15   | **3.25**    |
> |     | 8    | **3.40**   | **3.40**   | 8.24   | **3.31**  | 4.84  | **3.00**   |
> |    | 16   | 3.30  | **3.13** | 3.67   | **3.03**    | 3.68 | **2.80** |
> | **California**| 4    | 127695.29  | **124478.21** | 145197.65    | **121296.77**    | 175730.68     | **168763.81**  |
> |    | 8    | 122038.60   | **117945.30**   | 164740.07   | **115216.90**    | **165226.72** | 168103.52  |
> |      | 16   | 119206.11   | **115849.27**  | 156167.49  | **111100.62**  | **164411.93** | 169072.75      |
>
> **Results with variance**
> >We kindly remind you that we had provided the detailed results, including standard deviation, in Table 9 of the appendix in the original submitted version. We find that our ProtoLLM yields smaller variance in most cases, showing better stability.
>
> **full-shot results**
> >Following your suggestion, we report the full-shot results in  Table 4. As shown in Table 4, Our ProtoLLM with few-shot samples achieves the desired performance when compared to the full-shot performance, where the average performance of 64-shot is very close to the performance of full-shot. Besides, using our LLM-generated feature values on some datasets (Blood, Cultivars, Myocardial) in the few-shot scenario can even outperform the full-shot setting. This highlights the effectiveness of the LLM-generated feature values in improving performance even with limited data. We have added the full-shot performance in our revised version.
>
> ### Table 4: AUC scores comparison of ours with full-shot samples.
>
> | Dataset  | 0-shot | 4-shot | 8-shot | 16-shot | 32-shot | 64-shot | Full-Shot |
> |----|---|---|--|---|---|-----|----|
> | **Adult**      | 85.93  | 86.01  | 86.12  | 86.28   | 86.26   | 86.32   | 87.68 |
> | **Bank**       | 80.20  | 80.85  | 81.41  | 83.26   | 84.88   | 85.84   | 86.86 |
> | **Blood**      | 75.63  | 75.98  | 76.35 | 75.46   | 75.84   | 76.08   | 75.26     |
> | **Car**        | 78.29  | 79.41  | 80.40  | 82.22   | 84.78   | 87.45   | 92.44 |
> | **Credit-g**   | 61.29  | 62.25  | 63.26  | 64.52   | 68.32   | 71.75   | 73.44 |
> | **Cultivars**  | 58.93  | 59.37  | 60.51  | 60.45   | 60.63   | 61.67 | 58.83     |
> | **Diabetes**   | 75.65  | 75.68  | 75.76  | 75.70   | 77.48   | 78.00   | 82.18|
> | **Heart**      | 58.28  | 65.40  | 70.55  | 78.09   | 83.59   | 88.17   | 90.65 |
> | **Myocardial** | 62.52  | 63.25  | 63.62  | 64.03   | 65.44   | 65.75 | 65.50     |
> | **NHANES**     | 83.96  | 86.60  | 88.30  | 92.61   | 94.91   | 96.91   | 98.27 |
> | **Average**    | 72.07  | 73.48  | 74.63  | 76.26   | 78.21   | 79.79   | 81.11     |
>
> **truly novel datasets**
> >To generate the representative value for each feature, we query LLMs with detailed feature descriptions, this helps ProtoLLM to understand the novel features and thus has the ability to predict the correct values even for new features. Moreover, our ProtoLLM can be easily built upon different LLMs. As LLMs are updated iteratively, our ProtoLLM is able to understand newly created features. We are glad to test our ProtoLLM on novel features if the reviewer provides such datasets.

---

> ### Author Response · Authors · 2024-11-25
> **Thank you for your detailed and valuable review (3/3)**
>
> **There are existing works focused on feature generation**
> >We agree with you that feature generation is an efficient method for few-shot learning. Here we want to note that our method focuses on **feature value generation** using large language models, which is fundamentally different from **feature generation**. Feature generation creates new features, such as feature crossing or embeddings, to expand the feature set and is typically applied in scenarios with abundant data. In contrast, our method generates values for existing features without introducing new ones and is specifically designed for **few-shot tasks**, where data is scarce.
>
> **Comparison with LLM used Tabular learning frameworks**
> >Thank you for highlighting the importance of additional baselines. We have conducted experiments to compare our method with the approaches mentioned. The detailed results are provided in Table 5. Specifically, LIFT-ICL corresponds to the method from [1], while P2T is based on [2]. Unfortunately, as the code for [11] is not publicly available, we were unable to include it in our evaluation. As shown in this table, ours achieves the best performance on most of the datasets and achieves comparable performance on Diabetes, showing the effectiveness of ours.
>
> ### Table 5: AUC scores comparison with more baselines, where GPT-3.5 is used for all the baselines.
>
> | Dataset      | Shot | LIFT  | P2T   | ProtoLLM (ours) |
> |--------------|------|-------|-------|-----------------|
> | **Bank**     | 4    | 78.74  | 74.76  | **80.85**       |
> |              | 8    | **81.50** | 80.21  | 81.41          |
> |              | 16    | 80.32  | 80.14  | **83.26**       |
> | **Blood**    | 4    | 68.66  | 64.69  | **75.98**       |
> |              | 8    | 70.96  | 67.28  | **76.35**       |
> |              | 16    | 68.42  | 69.92  | **75.46**       |
> | **Credit-g** | 4    | 52.28  | 47.67  | **62.25**       |
> |              | 8    | 52.07  | 50.89  | **63.26**       |
> |              | 16    | 54.45  | 56.25  | **64.52**       |
> | **Diabetes** | 4    | **76.21** | 72.62  | 75.68          |
> |              | 8    | **78.82** | 71.15  | 75.76          |
> |              | 16    | 75.68  | 69.77  | **75.70**       |
> | **Average**  | 4    | 68.97  | 64.94  | **73.69**       |
> |              | 8    | 70.84  | 67.38  | **74.2**        |
> |              | 16    | 69.72  | 69.02  | **74.74**       |
>
> [1] LIFT: Language-Interfaced Fine-Tuning for Non-Language Machine Learning Tasks, NeurIPS 2022.
>
> [2] Tabular Transfer Learning via Prompting LLMs, COLM 2024.
>
> [3] Language models are weak learners, NeurIPS 2023.

---

> ### Comment · Reviewer_ToLN · 2024-11-28
> **Thank you for the rebuttal**
>
> I thank the author for their time and efforts in writing the rebuttal. I have read the updated paper, which I believe is much clearer and better written. Also, I appreciate additional experiments on multi-class datasets, regression, and comparisons with LLM-based methods. In the final revision, I think it would be great to add some baselines on full-shot just to highlight the current method's limitations for the future researchers. Furthermore, I think it is valuable to discuss the difference between feature generation in the related work in the revision. While only focusing on a few-shot can be seen as a limitation, I think this work is still valuable, and future works should work on more challenging scenarios (e.g., I hope to see LLMs surpass boosting tree methods on a full shot as well). Overall, I think it is a solid paper with a small limitation (e.g., the main focus is few-shot). Therefore, I have adjusted my score accordingly (to 6). I thank the authors again for their time and efforts.

---

> > ### Author Response · Authors · 2024-11-29
> > **Thank you for your reply**
> >
> > Thanks for your positive feedback and detailed suggestions, which are helpful for improving the quality of our submission.  We are glad that our response has addressed your concerns. We will follow your suggestions about related work and baselines on full-shot in the camera ready.

---

### Public Comment · ~Linlu_You1 · 2024-12-02
**Issues Report**

I suggest all the reviewers to consider the issues. First, I appreciate the authors for their interesting work. And I also appreciate all the reviewers for their efforts.
However, I find the reported results in this paper to be unconvincing.

1. **Prototype generation**. Since the key component of ProtoLLM are the constructed prototypes, the reasoning process of generating possible values for each class (Line 209 in revised paper) is important. However, for the example prompt shown in Fig. 3 and the corresponding response in Fig. 4, I observed significant instability in the responses generated by the LLM. Specifically, when I queried all the LLMs mentioned in the paper multiple times (K times) using the prompt from Fig. 3 as input, I was unable to replicate the response illustrated in the paper. Similar inconsistencies were also observed for the examples in Fig. 8 and 9.

2. **Reproducibility of AUC Results**. Based on this, I can not reproduce the reported AUC performance in the paper.

3. **The reported results for baseline methods exist issues**.
There are discrepancies in the experimental setups for baseline methods. As stated in the paper's 'Implementation Details' section, the results for STUNT, In-context Learning, TABLET, TabLLM, and FeatLLM were derived from [1]. While this is acceptable, the experimental protocol described in [1] differs from that of ProtoLLM. Specifically, in [1], few-shot samples for each class are generated by randomly selecting a certain number of samples (as per utils.py in [1]'s repository). In ProtoLLM, however, the authors use the first few samples in each class, as shown in data.py. Tabular data is highly sensitive to sampling, and this discrepancy could lead to significantly different results. To verify this, I applied ProtoLLM's protocol to FeatLLM and obtained different (and higher) results than those reported in the paper. This suggests an unfair comparison, especially given that FeatLLM is a primary baseline.

4. In addition, the reported results of LogReg, XGBoost, TabPFN are diiferent from that reported in [1].

5. **Missing values**. The authors remove columns with more than 20% missing values in data.py of the given code, which is not mentioned in the paper. According to [2], handling missing values is a critical aspect of feature heterogeneity. Since ProtoLLM aims to address heterogeneous tabular features, removing such columns oversimplifies the task and undermines its generalizability in the tabular domain.

6. **Performance Claims**. Solid performance is a key strength cited by many reviewers. However, given the issues with experimental protocols and reproducibility, the reported performance metrics are not sufficiently convincing. I suggest reviewers to carefully evaluate these concerns before considering a score increase.

7. **Uncertainty about few-shot definition**. The paper does not clearly define "few-shot" in the context of tabular learning. Does it refer to the total number of shots across all classes? If so, how does ProtoLLM perform with different numbers of classes, such as 3, 5, or 7? All experiments appear to use 0, 2, 4, 6, and 8 shots, leaving this aspect unexplored. It seems that the few-shot setting is quite different from that of N-way K-shot setting.

8. **Clarity of Writing**. Despite revisions, parts of the paper remain hard to read. For instance, Line 290 mentions 'From the Bayesian perspective,' but the Bayesian context is not adequately explained.

9. **Lack of Theoretical Support**. While the paper provides intuitive explanations for its motivation, it lacks robust theoretical support. Combined with the aforementioned issues regarding experimental results, this lack of theoretical grounding diminishes the credibility of the findings. Is there any theoretical support to demonstrate the feasibility of this method or clarify the scenarios in which it is applicable?

[1] Large language models can automatically engineer features for few-shot tabular learning.

[2] Deep Neural Networks and Tabular Data: A Survey.

---

> ### Author Response · Authors · 2024-12-03
> **Thank you for your interest in our approach (1/2)**
>
> First, we are glad that you are interested in our approach and point out your concerns about the reported results. For a clearer and more correct reproduction of our results, we have reorganized and updated our code (https://anonymous.4open.science/r/anonymous-2D04/protoLLM\_code.zip). Please check and refer to the ``readme.md`` file for step-by-step instructions. Besides, we also provide all generated features of 10 datasets in the ``oracle_features`` folder.
>
> **W1: Prototype generation**
> >
> >- We agree with you that the generated feature values are unstable due to the randomness brought by LLM. Notably, it is one of the reasons that we query the LLM multiple times. Besides, we also provided the variance in the results in the submitted version.
> >
> >- Figures 3-4 and 8-9 show the example responses rather than the only response from LLM. It makes sense that the feature values you reproduce are not exactly the same as in Figures 4 and 9. We suggest you reproduce the AUC score rather than the output feature value for more reasonable doubts.
> >- Possible inspection points. Please make sure that the version of LLM is ``gpt-3.5-turbo-0613`` and the used prompts. We rerun our code and find that the same answer in Fig.4 appears 14 times out of 30. We also tested ``GPT-4o mini`` and find that the same answer in Fig. 4 was generated 3 out of 10 times.
>
> **W2: Reproducibility of AUC Results**
> >Please carefully check out the version of LLM and the used prompts. You can refer to the ``oracle_features`` for the generated feature values or rerun ``run.py`` to generate values by yourselves. This makes sure the version of LLM and other hyperparameters remain the same as ours.
>
> **W3: The reported results for baseline methods exist issues**
> >- We kindly remind you that before using the first few samples in each class, we have shuffled all samples of each dataset (line 96 in ``data.py``). Therefore, our ProtoLLM also adopts a random sampling.
> >- You mentioned that you achieved better results with our protocol. We are curious about on which data this improvement is observed and by how much, and whether it falls within the range allowed by variance, as statistically, we consider it to be the same sampling strategy.
>
> **W4: the reported results of LogReg, XGBoost, TabPFN are diiferent from that reported in [1]**
> > Because we needed to conduct ablations (Fig. 7 in the submission) using the traditional approach, we reproduced these methods and reported their results by ourselves. We summarize the results reported by FeatLLM and those by ours in Table 1. We can observe that no matter using the reported results from FeatLLM [1] or the results reproduced by us, our proposed ProtoLLM always outperforms baselines. We will cite these results reported by [1] additionally in our camera ready if necessary.
>
> Table 1: Results of baselines in TfeatLLM and ProtoLLM. For each result in (a, b), a denotes the results in FeatLLM and b denotes the results in ProtoLLM.
> | Data           | Shot | LogReg        | XGBoost      | TabPFN       | ProtoLLM     |
> |----------------|------|---------------|--------------|--------------|--------------|
> | **average AUC**| 4    | (64.52, 60.80)| (50.00, 50.00)| (61.01, 61.96)| (74.62, 74.62)|
> |                | 8    | (70.76, 67.22)| (60.71, 60.42)| (66.98, 68.64)| (75.85, 75.85)|
> |                | 16   | (74.74, 72.22)| (68.72, 69.55)| (71.57, 72.55)| (77.62, 77.62)|
> |                | 32   | (78.02, 77.79)| (75.35, 74.47)| (75.42, 75.95)| (79.63, 79.63)|
> |                | 64   | (81.04, 81.40)| (78.42, 78.33)| (79.26, 79.74)| (81.35, 81.35)|
> | **average RANK**| 4    | (2.00, 2.6)   | (3.89, 3.7)   | (2.78, 2.44)  | (1.33, 1.1)   |
> |                | 8    | (2.00, 2.5)   | (3.56, 3.4)   | (3.00, 2.56)  | (1.44, 1.4)   |
> |                | 16   | (2.10, 2.5)   | (3.40, 3.3)   | (2.67, 2.33)  | (1.70, 1.6)   |
> |                | 32   | (2.20, 1.9)   | (3.10, 3.3)   | (2.67, 3.00)  | (1.90, 1.7)   |
> |                | 64   | (2.10, 2.0)   | (3.00, 3.3)   | (2.78, 2.56)  | (2.00, 2.0)   |
>
>
> **W5: Missing values**
> >For missing values, we follow FeatLLM [1] and other previous work [2-3] and simply handle them by removing the corresponding columns directly. We agree with you that dealing with missing values is a key aspect of feature heterogeneity, but this is beyond the scope of this paper.
>
>
> [1] Large Language Models Can Automatically Engineer Features for Few-Shot Tabular Learning. ICML 2024
>
> [2] The prevention and handling of the missing data, KJA 2013
>
> [3] When and how should multiple imputation be used for handling missing data in randomised clinical trials–a practical guide with flowcharts, BMC Medical Res. Methodol. 2017

---

> > ### Public Comment · ~Linlu_You1 · 2024-12-03
> >
> > Many thanks to the authors for providing their feedback. After carefully reviewing the response, and manuscript, I find that there are still significant issues that remain unresolved.
> >
> > Following the provided protocol, I attempted to reproduce the results shown in Fig. 4 using GPT-3.5-turbo-0613 and GPT-4o mini. However, I was unable to obtain the same outputs. Specifically, I am curious how the authors managed to achieve 14 occurrences out of 30 with GPT-3.5-turbo-0613 and 3 out of 10 with GPT-4o mini. This discrepancy raises concerns about reproducibility. I appeal to other reviewers and researchers to independently verify these results.
> >
> > While my failure to reproduce Fig. 4 might be due to coincidence, it also indicates that the proposed method may be experimentally unstable—a concern that the authors themselves acknowledge in their response. In the introduction (Lines 36–38), the authors emphasize the importance of tabular data in industries such as finance and healthcare. For methods intended for such critical application domains, experimental instability is a significant issue. It makes the method unreliable for practical use, especially in fields like finance or healthcare where robustness is crucial.
> >
> > It is understandable if a top-conference paper does not focus on practical deployment due to various reasons. However, instability in generated feature values is not an acceptable justification. If the work does not aim at practical applications, it should instead offer some theoretical guarantees to ensure the method’s reliability and utility. Unfortunately, the authors admit that they lack theoretical support, which leaves the reliability of the proposed method unsubstantiated. Throughout the paper, the explanations provided are purely intuitive. In my opinion, the method lacks both the experimental and theoretical rigor necessary for a top-tier conference.
> >
> > The authors claim that handling missing values is beyond the scope of the paper, and I agree that some methods are specifically designed for this. However, I did not require the authors to propose a method for handling missing values. My concern is that the authors did not rigorously clarify how missing values were addressed in their experiments. For example, recent benchmarks like [1] explicitly describe how missing values are handled. Yet, I could not find any such statement in the manuscript. Was there a reason the authors omitted this information?
> >
> > Furthermore, I disagree with the claim that removing columns with missing values is a common operation in the tabular domain. According to [2], a well-recognized benchmark for tabular data, none of the mainstream baselines directly remove columns with missing values. Instead, simple alternatives, such as filling with zero or the mean, are often employed. Therefore, the claim that handling missing values is beyond the scope of the paper does not seem reasonable. At the very least, the authors should clearly document whether columns with missing values were removed.
> >
> > In summary, this paper lacks sufficient experimental support (while it might be a coincidence that I cannot reproduce the same results as presented in the paper, this highlights potential instability that requires further verification by others). It also lacks theoretical support, a limitation acknowledged by the authors. Moreover, the writing does not appear rigorous, as evidenced by the insufficiently detailed discussion of missing values.  Although the authors might have promised to address these issues in the camera-ready version, the fact that such basic details are overlooked suggests that the paper is not yet ready for submission. Therefore, I believe the paper is not yet ready for a top-tier conference.
> >
> > [1] TALENT: A Tabular Analytics and Learning Toolbox
> >
> > [2] Revisiting Deep Learning Models for Tabular Data

---

> ### Author Response · Authors · 2024-12-03
> **Thank you for your interest in our approach (2/2)**
>
> **W7:Uncertainty about few-shot definition.**
>
> >Please see the baseline details of appendix A.4, where we stated For each random seed, 20% of the datasets are designated as the test set. We then perform a balanced sampling of K instances from the remaining data, following the methodology in [1, 4]. The relationship between the K used in ProtoLLM (denoted as KP rotoLLM ) and traditional N-way K-shot setting can be formulated as: $K_{ProtoLLM} = N × K$. During rebuttal, the reviewers suggested testing ProtoLLM on the TabZilla benchmark, and we found several datasets with an odd number of categories. Therefore, we used the N-way K-shot setting.
>
> **W9:Lack of Theoretical Support.**
> >This work provides a novel method to integrate the LLM into tabular few-shot classification, which we believe is interesting and useful for tabular learning. To sum up, our proposed method can be reproduced and we further provide the updated code for your convenience, where we provide more detailed descriptions for you. Our settings are generally following the closely-related work FeatLLM. Thanks for your time.
>
> To sum up, our proposed method can be reproduced and we further provide the updated code for your convenience, where we provide more detailed descriptions for you. Our settings are generally following the closely-related work FeatLLM. Thanks for your time.

---

> ### Author Response · Authors · 2024-12-03
> **Thank you for your interest in our approach**
>
> **Prototype generation**
> >To help you with how to prompt LLMs and reproduce the reasonable output, we provide our conversations with GPT-40 mini below (4/10 outputs the same values as Figure 4):
> >
> >https://chatgpt.com/share/674eea54-537c-8007-be2f-ed602b1040d4
> >
> >https://chatgpt.com/share/674eea9b-6bd8-8007-95b8-fab49068524a
> >
> >https://chatgpt.com/share/674eea8c-d7b4-8007-8b63-61c85b906849
> >
> >https://chatgpt.com/share/674ed472-65fc-8007-a08c-a5316290abd0
> >
> >https://chatgpt.com/share/674eea63-ea84-8007-b54f-f5649f0012e8
> >
> >https://chatgpt.com/share/674eea74-0db0-8007-b0e0-1f7cda093bef
> >
> >https://chatgpt.com/share/674ee58c-fecc-8007-bbbd-e1a06eb8124e
> >
> >https://chatgpt.com/share/674eeabd-8960-8007-9f27-f16d29f3fc31
> >
> >https://chatgpt.com/share/674eea40-5534-8007-b324-9a9011253618
> >
> >https://chatgpt.com/share/674ee61d-ee48-8007-915e-76a8b4d7321c
> >
> >Besides, we also want to note that Fig. 3-4 is an example of our ProtoLLM. It is reasonable that the feature values you reproduce are not exactly the same as in Figures 4 and 9. We find that we can produce high quality prototypes as long as the output of LLMs makes sense. We suggest you calculate the AUC score before making your comments.
>
> **instability**
> > We have reported the variance of our ProtoLLM in the appendix. The variance is a measure of model instability. Please take the time to check it.
>
> **Missing values**
> >As for missing values, we in this paper mainly focus on querying LLMs to prototype generation and thus follow previous works [1-3] (including FeatLLM) in dealing with missing values. We do not think this is a technical weakness of this paper.
>
> Finally, We thank all reviewers again for their time and valuable comments. The reviewers have provided our article with various high-quality suggestions. As experts in this field, I trust that they will make fair judgment.
>
> [1] Large Language Models Can Automatically Engineer Features for Few-Shot Tabular Learning. ICML 2024
>
> [2] The prevention and handling of the missing data, KJA 2013
>
> [3] When and how should multiple imputation be used for handling missing data in randomised clinical trials–a practical guide with flowcharts, BMC Medical Res. Methodol. 2017

---

> ### Author Response · Authors · 2024-12-03
> **Thank you for your interest in our approach**
>
> We respect that ICLR has an open review environment where anyone can comment to a paper. To help build a transparent and collaborative research community, we tried our best to response to your comments with detailed explanations and additional results, although we believe many of them are subjective, unfair and even incorrect. However, the quality of our paper is evaluated based on the professional reviews from our assigned reviewers, who have given professional and impartial comments to our paper. We trust that they will continue doing so without being biased by noises. We have decided not to response to you further. We thank you again for your interest in our paper and wish you have a nice day.

---

### Meta-Review · Area_Chair_6kcn · 2024-12-17

**Metareview:**

This paper proposes a new method for few-shot learning in tabular data using large language models (LLMs). Unlike existing approaches, this method avoids overfitting to the limited training examples by prompting the LLM without directly including them in the prompt. Instead, the few-shot samples are used to create "prototypes" that guide the LLM's predictions.

Strengths:
- The novel prompting strategy helps prevent overfitting to the small training sets.
- Additional experiments conducted during the rebuttal phase demonstrate the broader applicability of the method.

Weaknesses:
- The paper's writing is unclear and difficult to understand.
- The performance improvements over simpler baseline methods are minimal, despite the proposed method requiring significantly more computational resources for inference.
- The practical relevance of few-shot learning in tabular data remains unclear.

While the proposed method presents an interesting idea, the paper's weaknesses outweigh its strengths. The marginal performance gains, coupled with increased computational cost and unclear practical significance, lead me to recommend rejection.

**Additional Comments On Reviewer Discussion:**

The authors have significantly strengthened this paper through various experiments conducted during the rebuttal phase. These experiments demonstrate the versatility of their proposed method, showing its effectiveness in multi-class classification and regression tasks, as well as its compatibility with different LLMs. Incorporating these findings into the revised paper, along with improvements in clarity and writing, would greatly enhance its impact. Additionally, exploring the method's performance beyond few-shot settings could further increase its value and contribution to the field.

---

### Decision · Program_Chairs · 2025-01-22

Reject